# Task-induced neural covariability as a signature of approximate Bayesian learning and inference

**Richard D. Lange**[1,2¤]*, **Ralf M. Haefner**[1,2]*

**1** Brain and Cognitive Sciences, University of Rochester, Rochester, New York, United States of America,
**2** Center for Visual Science, University of Rochester, Rochester, New York, United States of America

¤ Current address: Department of Neurobiology, University of Pennsylvania, Philadelphia, Pennsylvania, United States of America
* lange.richard.d@gmail.com (RDL); ralf.haefner@rochester.edu (RMH)

**Data Availability Statement:** No new data were collected in this work. Matlab code for simulation results are available on https://github.com/haefnerlab/task-induced-noise-covariance.

## Abstract

Perception is often characterized computationally as an inference process in which uncertain or ambiguous sensory inputs are combined with prior expectations. Although behavioral studies have shown that observers can change their prior expectations in the context of a task, robust neural signatures of task-specific priors have been elusive. Here, we analytically derive such signatures under the general assumption that the responses of sensory neurons encode posterior beliefs that combine sensory inputs with task-specific expectations. Specifically, we derive predictions for the task-dependence of correlated neural variability and decision-related signals in sensory neurons. The qualitative aspects of our results are parameter-free and specific to the statistics of each task. The predictions for correlated variability also differ from predictions of classic feedforward models of sensory processing and are therefore a strong test of theories of hierarchical Bayesian inference in the brain. Importantly, we find that Bayesian learning predicts an increase in so-called "differential correlations" as the observer's internal model learns the stimulus distribution, and the observer's behavioral performance improves. This stands in contrast to classic feedforward encoding/decoding models of sensory processing, since such correlations are fundamentally information-limiting. We find support for our predictions in data from existing neurophysiological studies across a variety of tasks and brain areas. Finally, we show in simulation how measurements of sensory neural responses can reveal information about a subject's internal beliefs about the task. Taken together, our results reinterpret task-dependent sources of neural covariability as signatures of Bayesian inference and provide new insights into their cause and their function.

## Author summary

Perceptual decision-making has classically been studied in the context of feedforward encoding/ decoding models. Here, we derive predictions for the responses of sensory neurons under the assumption that the brain performs hierarchical Bayesian inference,

**Funding:** Funding was provided to RDL by an NIH traineeship (#T32 EY007125). RMH is funded by an R01 grant from the NIH (#5R01EY028811-04). The funders had no role in study design, data collection and analysis, decision to publish, or preparation of the manuscript.

**Competing interests:** The authors have declared that no competing interests exist.

including feedback signals that communicate task-specific prior expectations. Interestingly, those predictions stand in contrast to some of the conclusions drawn in the classic framework. In particular, we find that Bayesian learning predicts the increase of a type of correlated variability called "differential correlations" over the course of learning. Differential correlations limit information, and hence are seen as harmful in feedforward models. Since our results are also specific to the statistics of a given task, and since they hold under a wide class of theories about how Bayesian probabilities may be represented by neural responses, they constitute a strong test of the Bayesian Brain hypothesis. Our results can explain the task-dependence of correlated variability in prior studies and suggest a reason why these kinds of correlations are surprisingly common in empirical data. Interpreted in a probabilistic framework, correlated variability provides a window into an observer's task-related beliefs.

## Introduction

To compute our rich and generally accurate percepts from incomplete and noisy sensory data, the brain has to employ prior experience about which causes are most likely responsible for a given input [1, 2]. Mathematically, this process can be formalized as probabilistic inference in which posterior beliefs about the outside world (our percepts) are computed as the product of a likelihood function (based on sensory inputs) and prior expectations. These prior expectations reflect statistical regularities in the sensory inputs and contain information about the causes of those inputs and their relationships [3]. One approach to studying the neural basis of learning these regularities is to track them over the course of development. This approach was taken by Berkes et al (2011), who found signatures of learning a statistical model of sensory data in the changing relationship between evoked and spontaneous activity in visual cortex of ferrets [4]. A potential alternative approach exploits the experimenter's control of the regularities in the stimulus in the context of a psychophysical task. Such an approach could lead to a set of complementary predictions, different for each task, and comparable across tasks within the same brain. However, robust, task-specific, signatures of Bayesian learning are currently unknown, partly due to the fact that existing theories differ greatly in their assumptions for how probabilistic beliefs relate to tuning curves and correlated variability [3, 5–7].

Classic models frame perceptual decision-making as a signal-processing problem: sensory neurons transform input signals, and downstream areas separate task-relevant signals from noise [8]. Theoretical results based on this framework have shown how correlated variability among pairs of neurons impact both encoded information [9–17] as well as the correlations between sensory neurons and behavior ("choice probabilities"), that are used to quantify the involvement of individual neurons in a task [9, 18–21]. In particular, so-called "information-limiting," or "differential" correlations, which comprise neural variability in the same direction as neurons' tuning to some stimulus, have the greatest impact on both encoded information about that stimulus [16] and choice probabilities in a discrimination task [20]. These insights have motivated numerous experimental studies [20, 22–29] (reviewed in [30]). However, the extent to which choice probabilities and noise correlations are due to causally feedforward or feedback mechanisms is largely an open question [21, 24, 27, 31–35] that has profound implications for their computational role [30, 36–39]. Within the signal-processing framework, feedback signals are commonly conceptualized as endogenous attention that shapes neural tuning and covariability in such a way as to increase task-relevant information in the neural responses [40–42], reviewed in [43]. Therefore, one should expect differential correlations to

be reduced when a subject attends to task-relevant aspects of a stimulus. It is therefore notable that differential correlations have been empirically found to be among the few dominant modes of variance across a range of tasks and brain areas [26, 28, 29, 44]. From the classic signal-processing perspective, this is surprising because neural noise is usually assumed to be independent of task context, and to the extent that task-specific attention is engaged, it is expected to *decrease* these correlations. It is therefore puzzling that, when subjects switch between multiple tasks, correlations appear to dynamically re-align with the direction that maximally *limits* information in the current task [22, 27, 37] (but see [44]).

The Bayesian inference framework, on the other hand, premises that the goal of sensory systems is to infer the *latent causes* of sensory signals [1] (Fig 1). This has motivated models in which neural activity represents *distributions* of inferred variables [2, 45], reviewed in [3, 5, 6, 46, 47]. These models broadly fall into two categories that make specific assumptions about the relationship between probabilistic beliefs and neural responses: those inspired by Monte Carlo sampling algorithms [34, 48–55], and those inspired by parametric or variational algorithms [7, 45, 56–62]. Bayesian inference models interpret feedback connections in the brain as communicating contextual prior information or expectations [63–66]. While many neural models of Bayesian inference address neural (co)variability [34, 48, 52, 67–69], connections to the classic signal-processing framework, and to differential correlations and choice probabilities in particular, have been limited to the specific scope and assumptions of each model [34, 70, 71].

Here, we analytically derive connections between the two frameworks of Bayesian inference and classic signal-processing, while abstracting away from assumptions about a particular theory of distributional codes or the nature of the brain's internal model. Our key idea is that, while various theories of distributional coding may predict idiosyncratic neural signatures for the encoding of a *given* posterior distribution, the encoded posterior *itself* also changes trial-by-trial, and these changes in the encoded posterior must obey a self-consistency rule prescribed by Bayesian learning and inference. As we show, this self-consistency rule imposes a kind of symmetry between sensory neurons' sensitivity to an external stimulus and their sensitivity to changes in internal beliefs about the stimulus. We begin by reviewing concepts and introducing general notation for the relationship between the Bayesian inference and measured neural activity. We then turn to the specific case of discrimination tasks, analytically deriving that, after learning a task, optimal feedback of decision-related information should induce both differential correlations and choice probabilities aligned with neural sensitivity to the task's stimulus. These initial results require assumptions about noise and experimental design, but (in the low-noise case) are not tied to a particular theory of distributional codes, but hold for general distributional codes, including neural sampling [3], probabilistic population codes [58], distributed distributional codes [7], and others. The presence of non-negligible noise raises two additional considerations: first, we identify a novel class of "Linear Distributional Codes" (LDCs) for which our first set of results on decision-related feedback still hold; second, we show numerically that the constraint of self-consistent inference suppresses noise that is inconsistent with the task, leading to an apparent increase in differential correlations with learning. In total, our results contain two distinct normative mechanisms that predict apparent increases in differential correlations over learning, and one that predicts structure in choice probabilities. We then present initial evidence for these predictions in existing empirical studies, and suggest new experiments to test our predictions directly. Finally, we show in simulation how this theory allows the experimenter to glean information about the brain's beliefs about the task using only recordings from populations of sensory neurons.

## Results

### Distinguishing two sources of neural variability in distributional codes

A key challenge for Bayesian theories of sensory processing is linking observable quantities, such as stimuli, neural activity, and behavior, to hypothesized probabilistic computations which occur at a more abstract, computational level (Fig 1A and 1B). Following previous work, we assume that the brain has learned a generative model of its sensory inputs [3, 64, 73–75], and that populations of sensory neurons encode *posterior* beliefs over latent variables in the model conditioned on sensory observations: a hypothesis we refer to as "posterior coding." The responses of such neurons necessarily depend both on information from the sensory periphery, and on relevant information in the rest of the brain forming prior expectations. In a hierarchical model, likelihoods are computed based on feedforward signals from the periphery, and contextual expectations are relayed by feedback from other areas [64] (Fig 1B).

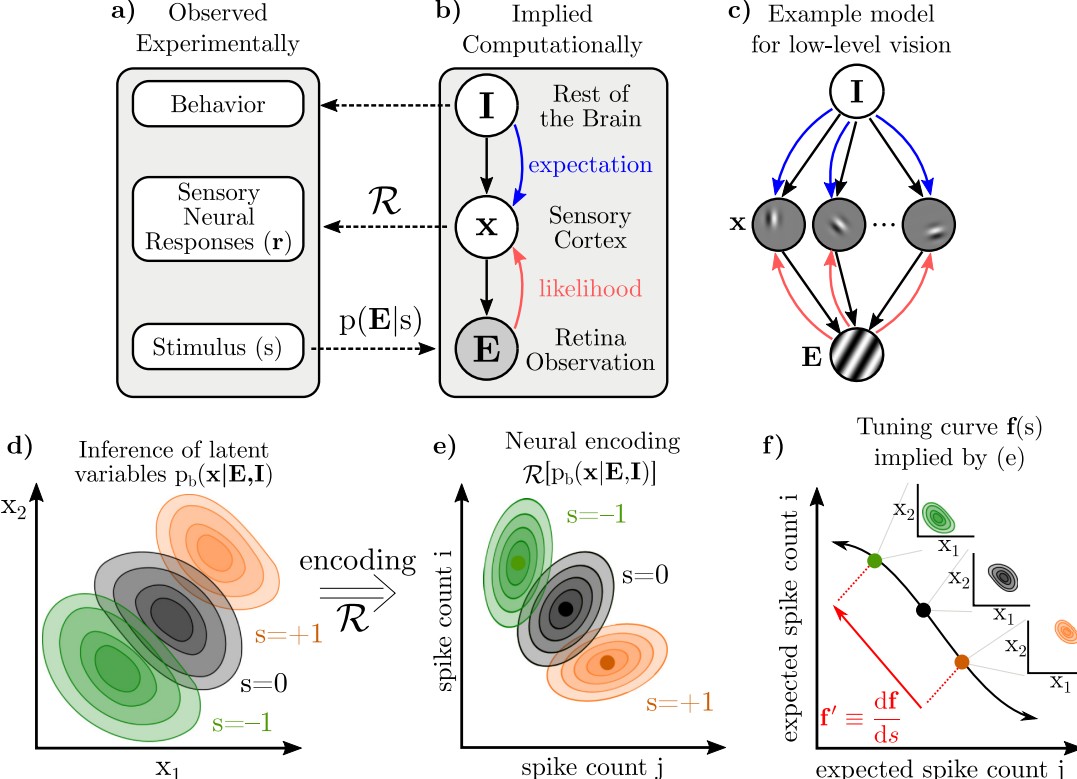

**Fig 1. Illustration of the components of our framework and how they relate to experimentally observed quantities. a-b)** The experimenter varies the sensory evidence, **E**, (e.g. images on the retina) according to *s* (e.g. orientation). The brain computes $p_b(\mathbf{x}|\mathbf{E}, \mathbf{I})$, its beliefs about latent variables of interest conditioned on those observations and other internal beliefs. **I** denotes these other "internal state" variables that are probabilistically related to, and hence form expectations for **x**. **c)** Here, we have illustrated **x** as Gabor patches which combine to form the image [72, 73], but our results hold independent of the nature of **x**. Solid black arrows represent statistical dependencies in the brain's implicit generative model, while red and blue lines show information flow. Dashed lines cross levels of abstraction. **d)** Varying a stimulus, *s*, such as the orientation of the grating image in (c), results in changes to the posterior over latent variables in the brain's internal model, but these distributions on **x** cannot be directly observed. **e)** The recorded neurons are assumed to encode the brain's posterior beliefs about **x** as in panel (d) through a distributional representation scheme, $\mathcal{R}$, which is a hypothesized map from unobservable distributions over **x** to observable distributions of **r**. We denote this map as $\mathcal{R}[p_b(\mathbf{x}|\mathbf{E}, \mathbf{I})]$ [45]. Our derivation initially assumes only that smoothly changing posteriors (d) correspond to smooth changes in neural response statistics (e); other restrictions on $\mathcal{R}$ will be introduced later. **f)** Mean spike counts (or firing rates) as a function of some stimulus *s* define a tuning curve, **f**(*s*). Both the tuning curve, **f**(*s*), and its tangent at a point, **f**, thus reflect, in part, changes in the underlying posterior over **x** (insets).

In our notation, **E** is the variable observed by the brain—the sensory input or evidence—and **x** is the (typically high-dimensional) set of latent variables. **I** is a high-dimensional vector representing all other internal variables in the brain that are probabilistically related to, and hence determine "expectations" for **x**. Note that the term "prior" is often overloaded, referring sometimes to stationary statistics learned over long time scales, and sometimes to dynamic changes in the posterior due to higher-level inferences or internal states. Therefore, we refer to the dynamic effect of internal states on **x** as "expectations". The brain's internal model of how these variables are related, $p_b(\mathbf{x}, \mathbf{E}, \mathbf{I})$, gives rise to a posterior belief, $p_b(\mathbf{x}|\mathbf{E}, \mathbf{I})$, that we assume to be represented by the recorded neural population under consideration. In a typical experiment, stimuli are parameterized by a scalar, $s$. For instance, when considering the responses of a population of neurons in primary visual cortex (V1) to a grating, $\mathbf{E}(s)$ is the grating image on the retina with orientation $s$, and **x** has been hypothesized to represent the presence or absence of Gabor-like features at particular retinotopic locations [76] or the intensity of such features [72, 77] (Fig 1B). We will return to this example throughout, but our results derived below are largely independent of the exact nature of **x**. In higher visual areas, for instance, **x** could be related to the features or identity of objects and faces [74, 75]. In these examples, **I** represents all other internal information that is relevant to **x**, such as task-relevant beliefs, knowledge about the context or visual surround, etc.

The rules of Bayesian inference allow us to derive expressions for structure in posterior distributions as the result of learning and inference. Importantly, the rules of probability apply to the relationships between abstract computational variables such as **E**, **x**, **I**, and their distributions, and *not* generally those between neural responses implementing those computations; it is a conceptually distinct step to link variability in posteriors to variability in neural responses encoding those posteriors. We use '$\mathcal{R}$' to denote the encoding from distributions over internal variables **x** into neural responses (Fig 1D) [45]. For reasonable encoding schemes $\mathcal{R}$, the chain rule from calculus applies: small changes in the encoded posterior result in small changes in the expected statistics of neural responses (Fig 1E, Methods). Using $f_i$ to denote the mean firing rate of neuron $i$, we can express its sensitivity to a change in stimulus, $s$, as

$$\frac{\mathrm{d}f_i}{\mathrm{d}s} = \left\langle \frac{\mathrm{d}f_i}{\mathrm{d}p_b(\mathbf{x}|\mathbf{E}(s))}, \frac{\mathrm{d}p_b(\mathbf{x}|\mathbf{E}(s))}{\mathrm{d}s} \right\rangle, \tag{1}$$

where $\langle \cdot, \cdot \rangle$ is an inner product in the space of distributions over **x**. In general, this expression should average over variability in the posterior due to sources other than $s$. For now, we are suppressing this extra "noise" for the sake of exposition, but will return to it later in Results. The second term in brackets is the change in the posterior as $s$ changes, and the first term relates those changes in the posterior to changes in the neuron's firing rate. Notice that Eq (1) bridges two levels of abstraction, from Bayesian inference (what happens to the posterior over **x** as $s$ is changed) to neural activity (how these changes to the posterior manifest as observable changes to neural activity).

It follows that, in the Bayesian inference framework, there are two distinct sources of neural variability acting at different levels of abstraction: variability in the encoding of a given posterior (Fig 2A–2C), and variability in the posterior itself (Fig 2D–2F) [78].

Distributional coding schemes [3, 5, 45, 47] typically assume that a given posterior may be realized in a distribution of possible neural responses, which we refer to as **variability in the encoding** (Fig 2A–2C). For instance, it has been hypothesized that neural activity encodes samples stochastically drawn from the posterior [34, 48–55, 69, 79]. Alternatively, neural activity may noisily encode fixed parameters of an approximate posterior [7, 58, 60–62, 80, 81]. Such distributional encoding schemes are reviewed in [3, 5, 6, 46, 47]. Previous

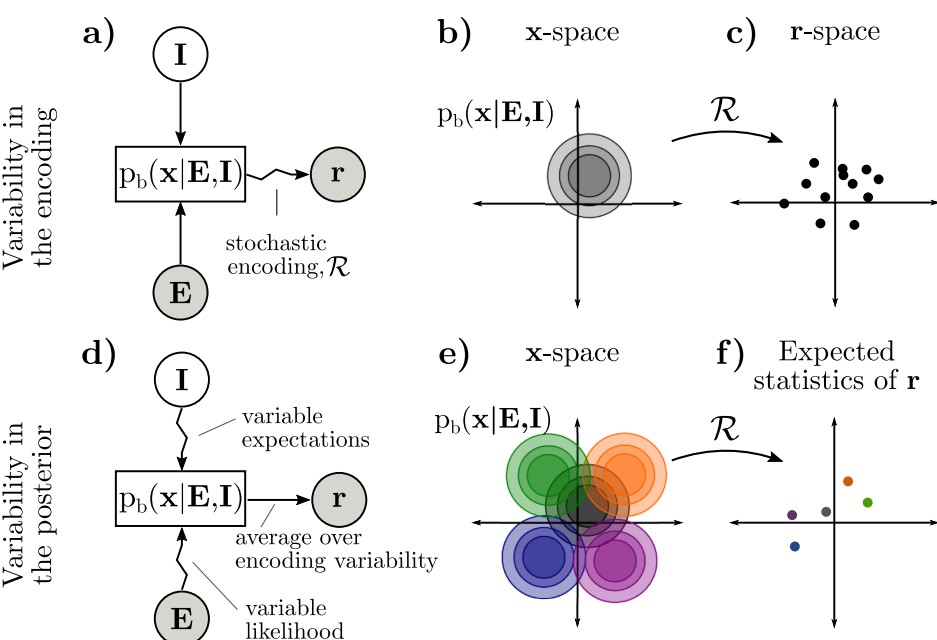

**Fig 2. Two distinct sources of neural co-variability in the Bayesian inference framework: Stochastic encoding of a fixed posterior (a-c) and variability in the posterior itself (d-f). a)** Consider the case where there is no variability in **I** or **E** and inference is exact (indicated by straight arrows from **E** and **I** to $p_b$), but posteriors are noisily realized in neural responses **r** (indicated by a zigzag arrow from $p_b$ to **r**). **b)** Exact inference always produces the same posterior for **x** for fixed **E** and **I**. **c)** The *neural encoding* of a given distribution may be stochastic, so a single posterior (b) becomes a distribution over neural responses **r**. The shape of this distribution may or may not relate to the shape of the posterior in (b), depending on the encoding (e.g. there is a correspondence in sampling, but not in parametric codes). **d)** Both sensory noise and variable expectations induce trial-to-trial variability in the posterior itself (indicated by zigzag arrows from **E** and **I** to $p_b$). This variability in the encoded posterior adds to variability in the encoding, as in (a-c), and can be understood as affecting the *average* or *expected* neural responses (indicated by a straight arrow from $p_b$ to **r**). **e)** Variability in the posterior can be thought of as a distribution of possible posteriors. **f)** Each individual posterior in (e) is a point in the space of expected statistics of **r**, such as expected spike counts. Variability in the underlying posterior may appear as correlated variability in spike counts.

work has linked (co)variability in neural responses to sampling-based encoding of the posterior [4, 34, 48, 52, 55, 67–69]. Our results are complementary to these; here we study trial-by-trial changes in the posterior itself, and how these changes affect the *expected statistics* of neural responses such as mean spike count and noise correlations of neural responses. Importantly, our results apply to a wide class of distributional codes including all of the above (Methods).

Trial-by-trial **variability in the encoded posterior** is an additional source of neural variability above and beyond variable encoding of a fixed posterior discussed above (Fig 2D–2F). There are two principal causes for variability in the posterior itself. First, there is variability in the observation **E**, or in neural signals relaying forward information about **E**, which induce variability in the likelihood [82–85]. Second, there is variability in internal states that may influence sensory expectations [37, 86]. Our initial results focus on the variability in the posterior due to variability in *task-relevant* beliefs or expectations [34, 86] since our primary goal is to understand task-specific influences on neural responses. Such variable expectations may reflect a stochastic approximate inference algorithm [48] or model mismatch, for example if the brain picks up on spurious dependencies in the environment as part of its model [78, 87–89]. Later, we will describe the effect of task-independent sources of variability in the posterior ("noise"), and how it is shaped by learning a task. In the remainder of this paper, we make

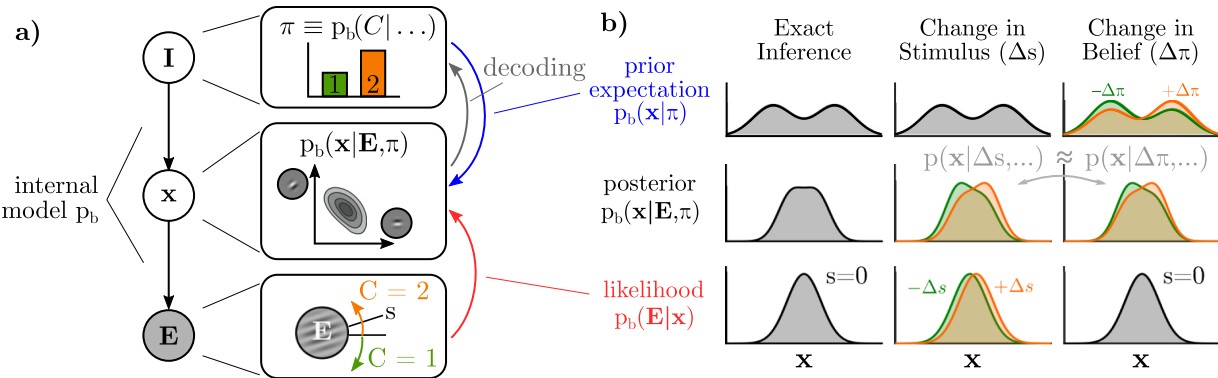

**Fig 3. a)** In a discrimination task, the brain performs inference over its latent variables ($p_b(\mathbf{x}|\ldots)$) and trial category (($p_b(C|\ldots)$)) conditioned on the sensory observation (**E**). We will first focus on the subject's graded belief about the binary category, written as $\pi \equiv p_b(C=1|\ldots)$, and ignore the influence of other internal states that are part of **I** (or assume they are fixed), before returning to them later. Implicitly, all inferences are with respect to an internal model $p_b$ (black arrows). A Bayesian observer learns a *joint* distribution between **x** and $C$, implying bi-directional influences during inference: $\mathbf{x} \to \pi$ is analogous to "decoding," while $\pi \to \mathbf{x}$ conveys task-relevant expectations. **b)** Visualization of how prior expectations (top row) and likelihood (bottom row) contribute to the posterior (middle row), with **x** as a one-dimensional variable. Changes to $s$ change the likelihood (middle column). Changes in expectation, $\pi$, imply changes in expectations (right column). Crucially, changes in the posterior in both cases (middle row) are approximately equal (gray arrow) as explained in the text.

these ideas explicit for the case of two-choice discrimination tasks for which much empirical data exists.

## Inference and discrimination with arbitrary sensory variables

In two-choice discrimination tasks, stimuli are classically parameterized along a single dimension, $s$, and subjects learn to make categorical judgments according to an experimenter-defined boundary which we assume to be at $s = 0$. We will use $C \in \{1, 2\}$ to denote the two categories, corresponding to $s < 0$ and $s > 0$. While our analytical results hold for discrimination tasks using other stimuli, for concreteness throughout this paper, our running example will be of orientation discrimination, in which case $s$ is the orientation of a grating with $s = 0$ corresponding to horizontal, and $C$ refers to clockwise or counter-clockwise tilts (Fig 3A). While our derivations make few assumptions (see Methods) about the nature of the brain's latent variables, **x**, our illustrations will use the example of oriented Gabor-like features in a generative model of images (Figs 1B and 3A).

Whereas much previous work on perceptual inference assumes that the brain explicitly infers relevant quantities defined by the experiment [2, 58, 60, 90], we emphasize the distinction between the external stimulus quantity being categorized, $s$, and the latent variables in the subject's internal model of the world, **x**. For the example of orientation discrimination, a grating image **E**($s$) is rendered to the screen with orientation $s$, from which V1 infers an explanation of the image as a combination of Gabor-like basis elements, **x**. The task of downstream areas of the brain—which have no direct access to **E** nor $s$—is to estimate the stimulus category based on a probabilistic representation of **x** (Fig 3A) [34, 91]. Crucially it is the posterior over **x**, rather than over $s$, which we hypothesize is represented by sensory neurons.

## Task-specific expectations and self-consistent inference

Probabilistic relations are inherently bi-directional: any variable that is predictive of another variable will, in turn, be at least partially predicted by that other variable. In the context of perceptual decision-making, this means that sensory variables, **x**, that inform the subjects' internal belief about the category, $C$, will be reciprocally influenced by the subject's belief about the

category (Fig 3A). Inference thus gives a normative account for feedback from "belief states" to sensory areas: changing beliefs about the trial category entail changing expectations about the sensory variables whenever those sensory variables are part of the process of forming those categorical beliefs in the first place [34, 64, 86, 92].

A well-known self-consistency identity for probabilistic models is that their prior is equal to their average inferred posterior, assuming that the model has learned the data distribution ([3, 4, 93]; Methods). We can use this identity to write an expression for the optimal prior over $\mathbf{x}$ upon learning the statistics of a task (Methods):

$$p_b(\mathbf{x}|C = c) = \mathbb{E}_{p_e(s|C=c)}[p_b(\mathbf{x}|\mathbf{E}(s))].\tag{2}$$

Eq (2) states that, given knowledge of an upcoming stimulus' category, $C = c$, the optimal prior on $\mathbf{x}$ is the average posterior from earlier trials in the same category [94].

Throughout, we use the subscript 'b' to refer to the brain's internal model, and the subscript 'e' to refer to the experimenter-defined model (Methods). To use the orientation discrimination example, knowing that the stimulus category is "clockwise" increases the expectation that clockwise-tilted Gabor features will best explain the image, *since they were inferred to be present in earlier clockwise trials*. Importantly, Eq (2) is true regardless of the nature of $\mathbf{x}$ or $s$. It is a *self-consistency* rule between prior expectations and posterior inferences that is the result of the brain having fully learned the statistics of its sensory inputs in the task, i.e. $p_b(\mathbf{E}) = p_e(\mathbf{E})$ ([3, 4, 93]; Methods). This self-consistency rule allows us to relate neural responses due to the stimulus ($s$) to neural responses due to internal beliefs ($\pi$) without specific assumptions about $\mathbf{x}$.

In binary discrimination tasks, the subject's belief about the correct category is a scalar quantity, which we denote by $\pi = p_b(C = 1|\ldots)$. Given $\pi$, the optimal expectations for $\mathbf{x}$ are a correspondingly graded mixture of the per-category priors:

$$p_b(\mathbf{x}|\pi) = \pi p_b(\mathbf{x}|C = 1) + (1 - \pi)p_b(\mathbf{x}|C = 2).\tag{3}$$

The posterior over $\mathbf{x}$ for a single trial depends on both the stimulus and belief *for that trial*:

$$p_b(\mathbf{x}|\pi, \mathbf{E}(s)) \propto p_b(\mathbf{E}(s)|\mathbf{x})p_b(\mathbf{x}|\pi).\tag{4}$$

We assume beliefs about $C$ (i.e. $\pi$) and $\mathbf{x}$ are represented by separate neural populations, for instance if $\pi$ is represented in a putative decision-making area of the brain while $\mathbf{x}$ is in sensory cortex; however, their relation to each other via anatomically "feedforward" or "feedback" pathways is not crucial for the main result except in the context of designing specific experimental interventions. For simplicity, we will say task-related expectations "feed back" to $\mathbf{x}$, as this is the common and familiar case in which $\mathbf{x}$ stands for whatever is represented by sensory cortex. We will next derive the specific pattern of neural correlated variability when $\pi$ varies.

### Variability in the posterior due to changing task-related expectations

Even when the stimulus is fixed, subjects' beliefs and decisions are known to vary [8]. Small changes in a Bayesian observer's categorical belief ($\Delta\pi$) result in small changes in their posterior distribution over $\mathbf{x}$, which can be expressed as the derivative of the posterior with respect to $\pi$:

$$\left.\frac{d}{d\pi}p_b(\mathbf{x}|\mathbf{E}(s = 0), \pi)\right|_{\pi = 1/2},$$

where $s = 0$ and $\pi = {}^1/_2$ indicate that the derivative is taken at the category boundary where an unbiased observer's belief is ambivalent. Note that a 50/50 prior at $s = 0$ is not required, and

our results would still hold after replacing all instances of $\pi = {}^1/_2$ with whatever the observer's belief is at $s = 0$.

Our first result is that this derivative is *approximately proportional* to the derivative of the posterior with respect to the stimulus. Mathematically, the result is as follows:

$$\frac{d}{d\pi} p_b(\mathbf{x}|\mathbf{E}(s=0), \pi)\bigg|_{\pi={}^1/_2} \tilde{\propto} \frac{d}{ds} p_b(\mathbf{x}|\mathbf{E}(s), \pi = {}^1/_2)\bigg|_{s=0}, \tag{5}$$

where the symbol $\tilde{\propto}$ should be read as "approximately proportional to" (see Methods for proof). This result is visualized in one dimension in Fig 3B: small changes in categorical expectation ($\pm\Delta\pi$) and small changes in the stimulus ($\pm\Delta s$) result in strikingly similar changes to the posterior over $\mathbf{x}$.

Eq (5) states that, for a Bayesian observer, small variations in the stimulus around the category boundary have the same effect on the inferred posterior over $\mathbf{x}$ as small variations in their categorical beliefs. The proof makes four assumptions: first, the subject must have fully learned the task statistics, as specified by equations (2) and (3). Second, the two stimulus categories must be close together, i.e. the task must be near or below psychometric thresholds. Third, variations of stimuli within each category must be small.

We further discuss these conditions and possible relaxations in S1 Text, and visualize them in S1 and S2 Figs. Finally, we have assumed that there are no additional noise sources causing the posterior to vary; we consider the case of noise in the section "Effects of task-independent noise" below.

## Feedback of variable beliefs implies differential correlations

"Information-limiting" or "differential" correlations refer to neural (co)variability that is indistinguishable from changes to the stimulus itself, i.e. variability in the $\mathbf{f}' \equiv d\mathbf{f}/ds$ direction for a fixed $s$. For a pair of neurons $i$ and $j$, differential *covariance* is proportional to $f_i' f_j'$ [16]. Covariance is transformed to correlation by dividing by the square root of the product of both neurons' variances, $\sigma_i \sigma_j$, which gives an expression for differential *correlation* proportional to $d_i' d_j'$, where $d_i' \equiv f_i'/\sigma_i$ is the normalized "d-prime" neurometric sensitivity of neuron $i$ [95].

Applying the "chain rule" in Eq (1) to Eq (5), it directly follows that

$$\frac{d\mathbf{f}}{d\pi}\bigg|_{\substack{s=0 \\ \pi={}^1/_2}} \tilde{\propto} \frac{d\mathbf{f}}{ds}\bigg|_{\substack{s=0 \\ \pi={}^1/_2}}, \tag{6}$$

implying that the effect of small changes in the subject's categorical beliefs ($\pi$) is approximately proportional to the effect of small changes in the stimulus on the responses of sensory neurons that encode the posterior. Both induce changes to the mean rate in the $\mathbf{f}' \equiv d\mathbf{f}/ds$ direction.

Importantly, when a subject is trained on multiple tasks and given a cue about the task context, the influence of their categorical expectations on posterior-coding neurons should then depend on the cue, since the cue informs their expectations of possible stimuli. Below, this task-dependence of $\mathbf{f}'$ will enable strong experimental tests of this theory using a cued task-switching paradigm.

A direct consequence of Eq (5) is that variability in $\pi$ adds to neural covariability in the $\mathbf{f}'$–direction above and beyond whatever intrinsic covariability was present before learning. We obtain, to a first approximation, the following expression for the noise covariance between neurons $i$ and $j$:

$$\Sigma_{ij} = \Sigma_{ij}^{\text{intrinsic}} + \Sigma_{ij}^{\text{belief}}, \tag{7}$$

where $\Sigma^{\text{intrinsic}}$ captures "intrinsic" noise such as Poisson noise in the encoding. It follows from (6) that

$$\Sigma_{ij}^{\text{belief}} \overset{\sim}{\propto} \text{var}(\pi)\mathbf{f}_i'\mathbf{f}_j'^{\top} . \tag{8}$$

Interestingly, this is exactly the form of so-called "information-limiting" or "differential" covariability [16]. In the feedforward framework, differential covariability arises due to afferent sensory or neural noise and limits the information about $s$ in the population [16, 30, 33]. This source of differential correlations is captured by $\Sigma^{\text{intrinsic}}$ and is assumed to be always present, independent of task context. Here, we highlight an additional source of differential correlations due to feedback of variable beliefs about the stimulus category. Unless these beliefs are *true*, or unless downstream areas have access to and can compensate for $\pi$, the differential covariability induced by $\pi$ limits information like its bottom-up counterpart ([27, 30, 37]; also see Discussion). Importantly, and unlike the feedforward component of differential covariability, the feedback differential covariability predicted here arises as the result of task-learning and is predicted to *increase* while behavioral performance in the task improves, and to *change* depending on the cued task context in a task-switching paradigm.

## Feedback of variable beliefs implies choice probabilities aligned with stimulus sensitivity

A direct prediction of the feedback of beliefs $\pi$ to sensory areas, according to (6), is that the average neural response preceding choice 2 will be biased in the $+\mathbf{f}'$ direction, and the average neural response preceding choice 1 will be biased in the $-\mathbf{f}'$ direction, since the subject's behavioral responses will be based on their belief, $\pi$. Excess variability in $\pi$, e.g. due to spurious serial dependencies, or simply approximate inference, will therefore introduce additional correlations between neural responses and choices above and beyond those predicted by a purely feedforward "readout" of the sensory neural responses [8, 19, 20, 31, 32, 34, 96]. Our results predict that this top-down component of choice probability should be proportional to neural sensitivity:

$$\text{CP}_i - \frac{1}{2} \overset{\sim}{\propto} d_i', \tag{9}$$

where $d_i' \equiv f_i'/\sigma_i$ is the "d-prime" sensitivity measure of neuron $i$ from signal detection theory [95] (Methods). Interestingly, the classic feedforward framework makes the same prediction for the relation between neural sensitivity and choice probability assuming an optimal linear decoder [19], raising the possibility that, contrary to previous conclusions, the empirically observed relationship between CPs and neural sensitivity that emerges over learning [97] is not just due to changes in the feedforward read-out as commonly assumed [8, 98] but is instead the result of changes in feedback indicating variable beliefs.

## Effects of task-independent noise

The above results assumed no measurement noise nor variability in other internal states besides the relevant belief $\pi$. Just as in the previous sections, we will distinguish variability *in the encoded posterior*, to which we can apply the self-consistency constraint of Bayesian inference, from neural variability due to the neural representation of a fixed posterior through $\mathcal{R}$. In the presence of noise, the posterior itself changes from trial to trial even for a fixed stimulus $s$ and fixed beliefs $\pi$ [83]. To study the consequences of this added variability, we introduce a variable, $\boldsymbol{\epsilon}$, that encompasses all sources of task-independent noise on each trial, and condition the posterior on its value: $p(\mathbf{x}|E(s), \pi; \boldsymbol{\epsilon})$ (Methods). Following our decomposition of variance

in [Fig 2D](), this noise, like $\pi$, acts *on the encoded posterior* rather than on the neural responses directly. Unlike $\pi$, $\boldsymbol{\epsilon}$ is by definition task-independent ([Methods]()). This impacts our initial results in two principal ways, laid out in the following two sections: first, although ideal learning and self-consistent inference still implies that the average posterior equals the prior ([Eq (2)]()), the average must now be taken over both $s$ and the distribution of noise p($\boldsymbol{\epsilon}$). This reduces the generality of our main result on the proportionality of $\frac{df}{ds}$ and $\frac{df}{d\pi}$ ([Eq (6)]()), but leads us to consider a new classification of *linear* distributional codes (LDCs) with interesting robustness properties to noise. Second, task-independent noise can be amplified or suppressed by a task-dependent prior, which we also find to have a task-dependent effect on neural covariability that sometimes, but not always, increases differential correlations.

**Variable beliefs in the presence of noise.**   Previously, we decomposed a neuron's sensitivity to the stimulus into the product of its sensitivity to changes in the encoded posterior with changes in the posterior due to the stimulus. In the presence of noise, $\frac{df_i}{ds}$ now requires averaging over realizations of the posterior for different values of the noise, $\boldsymbol{\epsilon}$. On the other hand, we previously saw that a neuron's sensitivity to feedback of beliefs, $\frac{df_i}{d\pi}$, depends on the sensitivity of $f_i$ to the *average posterior*. This distinction between the *average encoding of posteriors*, which defines $\frac{df_i}{ds}$, and the *encoding of the average posterior*, which defines $\frac{df_i}{d\pi}$, is crucial. In general, the expected value of a nonlinear function is not equal to that function of the expected value, hence the alignment between the two vectors, $\frac{df}{ds}$ and $\frac{df}{d\pi}$, no longer holds in general ([Methods]()).

However, there is a special class of encoding schemes in which firing rates are linear with respect to *mixtures* of distributions over **x**. We call these **Linear Distributional Codes** (LDCs). For LDCs, the two expectations mentioned above become identical, and we recover our earlier results for both task-dependent noise covariance ([Eq (8)]()) and structured choice probabilities ([Eq (9)]()) ([Methods]()). Examples of LDCs include sampling codes where samples are linearly related to firing rate [34, 48–51, 91] as well as parametric codes where firing rates are proportional to expected statistics of the distribution [7, 45, 56, 99]. Examples of distributional codes that are *not* LDCs include sampling codes with a nonlinear relationship between samples and firing rates [52, 53, 55], as well as parametric codes in which the *natural parameters* of an exponential family are encoded [58, 60, 62, 80].

This suggests a novel method for experimentally distinguishing between large classes of models: if, across multiple task contexts, manipulating the amount of external noise results in no changes to the axis along which feedback modulates sensory neurons, then this suggests that the brain's distributional code may be linear.

**Interactions between task-independent noise and task-dependent priors.**   Although we assumed that noise $\boldsymbol{\epsilon}$ arises from task-independent mechanisms, it is nonetheless shaped by task learning: task-independent noise in the likelihood interacts with a task-specific prior to shape variability in the posterior ([Fig 4]()). Below, we show that this mechanism can change task-dependent differential correlations in neural responses even if a subject's beliefs ($\pi$) do not vary. This idea is reminiscent of recent mechanistic models of neural covariability in which a static internal state, such as attention to a location in space, influences the dynamics of a recurrent circuit, allowing variation in some directions while suppressing others [100, 101]. Here, we investigate what happens when the brain employs a static prior over stimuli for a given task, finding that noise is selectively suppressed in certain directions by the prior.

We again study the trial-by-trial variability in the posterior itself as opposed to the shape or moments of the posterior on any given trial. Formally, we study the covariance due to noise ($\boldsymbol{\epsilon}$) in the posterior *density* at all pairs of points $\mathbf{x}_i$, $\mathbf{x}_j$, i.e. $\Sigma_{ij} \equiv \mathrm{cov}(p_b(\mathbf{x}_i|\ldots), p_b(\mathbf{x}_j|\ldots))$. We have shown ([Methods]()) that, to a first approximation, the posterior covariance is given by a product of the covariance of the task-independent noise in the likelihood, $\Sigma^{\mathrm{LH}}(\mathbf{x}_i, \mathbf{x}_j)$, and the

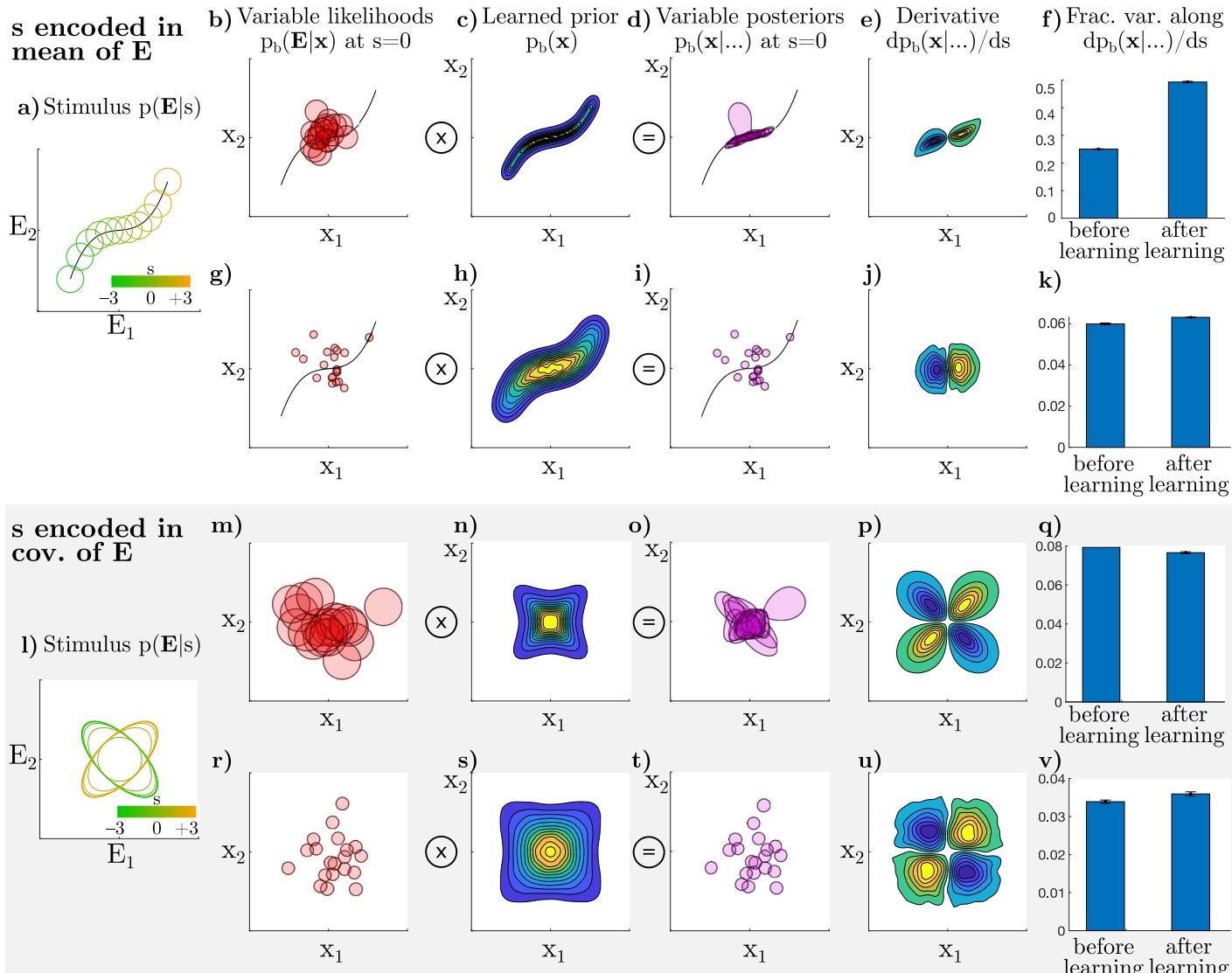

**Fig 4.** Numerical simulation of how variable likelihoods both determine and interact with the shape of the prior. Top two rows: mean of observations **E** depends on *s* as shown in panel (a). Bottom two rows (gray background): covariance of observations **E** depends on *s* as shown in panel (l). **a)** In this setup, observations are noisy, reflecting both internal and external noise. We set **E** to be two-dimensional, drawn from $p_e(\mathbf{E}|s)$, a Gaussian whose mean depends on *s* (black curve). Ellipses indicate one standard deviation of $p_e(\mathbf{E}|s)$. **b)** Now visualizing the space of latent variables, **x**, which we also set to be two-dimensional, such that the brain's internal model assumes $p_b(\mathbf{E}|\mathbf{x})$ is Gaussian, centered on **x**, with the shown variance. Each red circle shows a single contour of the likelihood, as a function of **x**, for a different **E** drawn from the $p_e(\mathbf{E}|s = 0)$ set of observations. **c)** After learning, the prior is extended along the curve that parameterizes the mean of **x**, such that the brain's distribution on observations, $p_b(\mathbf{E})$ marginalized over **x**, approximates the true distribution on observations, $p_e(\mathbf{E})$ marginalized over *s* (Methods). **d)** Posteriors in the zero-signal or *s* = 0 case, given by the product of the likelihoods in (b) with the prior in (c). **e)** The direction in distribution-space corresponding to differential covariance in neural-space is the $dp_b(\mathbf{x})/ds$-direction, averaged over instances of noise. **f)** The fraction of variance in distribution-space (d) along the $dp_b(\mathbf{x})/ds$-direction. After learning, a larger fraction of the total variance is in the $dp_b(\mathbf{x})/ds$-direction, corresponding to increased differential correlations in neural space. **g-k)** Identical to (b) through (f), except that the brain's internal model is more precise relative to the variations in observations **E**, modeled as smaller variance for $p_b(\mathbf{E}|\mathbf{x})$. This has the effect of reducing the effect of learning a new prior in panel (k). **l-v)** Identical to (a) through (k), except that here, *s* parameterizes the *covariance* of observations **E** rather than their mean. This is analogous to, for example, an orientation-discrimination task with randomized phases, such that the *average* stimulus in pixel-space is identical for both categories. While panel (v) (precise model) shows a slight increase in variance in the $dp_b(\mathbf{x}|...)/ds$ direction, consistent with (f) and (k), panel (q) (imprecise model) shows a slight *decrease*.

brain's prior over $\mathbf{x}_i$ and $\mathbf{x}_j$:

$$\Sigma(\mathbf{x}_i, \mathbf{x}_j) \propto p_b(\mathbf{x}_i) \Sigma^{\mathrm{LH}}(\mathbf{x}_i, \mathbf{x}_j) p_b(\mathbf{x}_j) . \tag{10}$$

The effect of learning a task-dependent prior in Eq (10) can be understood as "filtering" the noise, suppressing or promoting certain directions of variability in the space of posterior distributions.

Differential correlations emerge from this process if variability in the $dp_b(\mathbf{x}|...)/ds$-direction is less suppressed than in other directions. Whether this is the case, and to what extent, depends on the interaction of $s$ and $\mathbf{x}$, an analytic treatment of which we leave for future work. Here, we present the results from two sets of numerical simulations: one in which the *mean* of $\mathbf{x}$ depends on $s$ (Fig 4A–4K) and one in which the *covariance* of $\mathbf{x}$ depends on $s$ (Fig 4L–4V).

In these simulations, we assume both $\mathbf{x}$ and $\mathbf{E}$ to be two-dimensional, with isotropic Gaussian likelihoods over $\mathbf{E}$ given $\mathbf{x}$ in the brain's internal model. To connect back to our earlier examples of visual discrimination tasks, this two-dimensional stimulus space can be thought of as a simplified two-pixel image. We fixed the brain's internal likelihood $p_b(\mathbf{E}|\mathbf{x})$ over learning, and modeled the learning process as gradient descent of a loss function that encourages the brain's model of observations to approach their true distribution, i.e. changing the prior $p_b(\mathbf{x})$ so that the marginal likelihood $p_b(\mathbf{E})$ moves towards the true distribution of stimuli in each experiment $p_e(\mathbf{E})$ (Methods). Importantly, learning such a correspondence between $p_e(\mathbf{E})$ and $p_b(\mathbf{E})$ in the stimulus space is a sufficient condition for self-consistent beliefs about one's internal latent variables, $\mathbf{x}$ [4]. In these simulations, the brain's posterior $p_b(\mathbf{x}|...)$ varies for a fixed $s$ because different values of $\mathbf{E}$ are provided for the same $s$; this variability in $\mathbf{E}$ given $s$ is the result of both external and internal noise sources. In the first set of simulations, the *mean* of the observations depended on $s$ (Fig 4A). Small variations in $s$ around the boundary $s = 0$ primarily translated the posterior, resulting in a two-lobed $dp_b(\mathbf{x}|...)/ds$ structure (Fig 4E and 4J). After learning, the prior sculpted the noise such that trial-by-trial variance in posterior densities was dominated by translations in the $dp_b(\mathbf{x}|...)/ds$-direction (Fig 4F and 4K).

The intuition behind this first set of results is as follows. During learning, both uninformative $s = 0$ and informative $s < 0$ or $s > 0$ stimuli are shown. As a result, the learned prior (equal to the average posterior) becomes elongated along the curve that defines the mean of the likelihood (Fig 4C and 4H), which is also the direction that defines $dp_b(\mathbf{x}|...)/ds$ (Fig 4E and 4J). After learning, if noise in $\mathbf{E}$ happens to shift the likelihood along this curve, then the resulting posterior will remain close to that likelihood because the prior remains relatively flat along that direction. In contrast, noise that changes the likelihood in an orthogonal direction will be "pulled" back towards the prior (Fig 4B–4D). Thus, multiplication with the prior preferentially suppresses noise orthogonal to $dp_b(\mathbf{x}|...)/ds$. Applying the chain rule from Eq (1), this directly translates to privileged variance in the differential or $\mathbf{f}'\mathbf{f}'^\top$ direction in neural space. As shown in Fig 4G–4K, the magnitude of this effect further depends on the width of the brain's likelihood relative to the width of the prior. We find that decreasing the variance (increasing the precision) of $p_b(\mathbf{E}|\mathbf{x})$ dramatically attenuates the change (Fig 4K). Intuitively, this is because any effect of changing the prior is less apparent when the likelihood is narrow relative to the prior. Importantly, both scenarios occur in common perceptual decision-making tasks [102].

While these first results are intuitive, they rely, in part, on the assumption that the primary effect of $s$ is to translate the posterior on $\mathbf{x}$. In a second set of simulations, we investigated what happens when $s$ determines the covariance of $\mathbf{E}$ rather than its mean (Fig 4L–V). This is analogous to an orientation discrimination task with randomized phases, since the mean image is identical in both categories, so the category is determined by coordinated changes in "pixels." Otherwise, the brain's internal model, the learning procedure, and noise were identical to the

first set of simulations. In the case of relatively narrow likelihoods, we again found that the fraction of variance in the $dp_b(\mathbf{x}|\ldots)/ds$-direction slightly increased after learning (Fig 4V), consistent with the first set of results. Surprisingly, we found the reverse effect—a slight *decrease* in the fraction of variance in the $dp_b(\mathbf{x}|\ldots)/ds$-direction, when the brain's likelihood was relatively wide (Fig 4Q). While the first set of results—where *s* primarily translates the posterior on **x**—appear robust, this second set of results indicates that the interaction between Bayesian learning and noise is subtle, and whether it results in an increase or decrease in the (relative) variability along the **f′** direction in neural space depends on the particular relationship between *s*, **E**, and **x**. We leave a further exploration of this interaction to future work.

Note that whereas our results on variability due to the feedback of variable beliefs implied an increase in neural *covariance* along the $\mathbf{f}'\mathbf{f}'^{\top}$-direction over learning, the effect of "filtering" the noise does not necessarily increase nor decrease variance, even in cases where the *fraction* of variance along $dp_b(\mathbf{x}|\ldots)/ds$ increases. Adapting a prior to the task may affect both total and relative amounts of variance in the task-relevant direction, depending on the brain's prior at the initial stage of learning.

## Connections with empirical literature

To summarize, we have identified three signatures of Bayesian learning and inference that should appear in neurons that encode the posterior over sensory variables: (i) a feedback component of choice probabilities proportional to normalized neural sensitivity (Eq (9)), (ii) a feedback component of noise covariance in the "differential" or $\mathbf{f}'\mathbf{f}'^{\top}$ direction (Eq (8)), and (iii) additional structure in noise correlations due to the "filtering" of task-independent noise by a task-dependent prior. We emphasize that our results only describe how learning a task *changes* these quantities, and makes no predictions about their structure before learning. In this section, we highlight previous literature that has isolated *changes* in choice probabilities and noise correlations associated with a particular task, whether over learning or due to task-switching.

Eq (9) predicts that the top-down component of choice probability should be proportional to the vector of normalized neural sensitivities to the stimulus.

Indeed, such a relationship between CP and d′ was found by many studies (reviewed in [96, 103, 104]). For instance, Law and Gold (2008) demonstrated a relationship between neurometric sensitivity (quantified by neurometric threshold, proportional to 1/d′) and choice probability that only emerged after extensive learning. However, this finding can also be explained in a purely feedforward framework assuming an optimal linear decoder [19, 98]. Thus, the empirical finding that choice probabilities are often higher for neurons with higher neurometric sensitivity supports both the theories of feedforward optimal linear decoding and our theory based on feedback of a categorical belief, and in general CP will reflect a mix of both feedforward and feedback mechanisms [35]. The relative importance of these two mechanisms in producing CPs remains an empirical question. For much stronger support for our theory, we will therefore focus on empirical data on noise correlations, whose task-dependence is much harder to explain using feedforward mechanisms, and for which no feedforward explanation has been proposed.

Two existing studies have isolated task-dependent component of noise covariance by holding the stimulus constant while switching between two comparable tasks that a subject is performing, thus altering their task-specific expectations while minimally affecting the feedforward drive. The difference in neural response statistics to a stimulus that is *shared by both tasks* isolates the task-dependent component to which our results apply. Bondy et al [27] recorded from neural populations in macaque V1 while the monkeys switched between

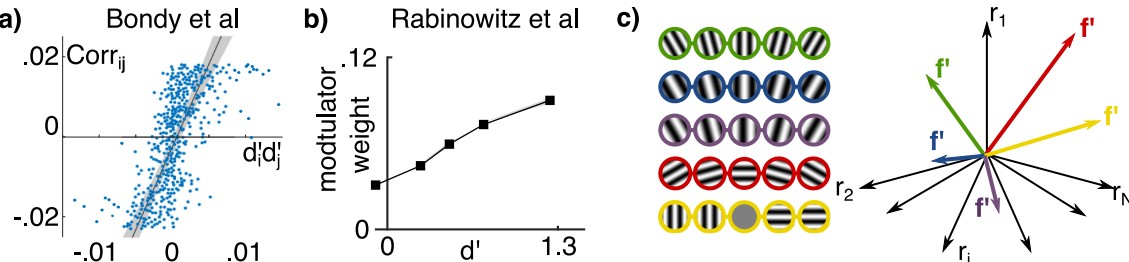

**Fig 5. Examples of predicted effects in existing empirical literature.** Corr denotes correlation, and $d'_i$ is the normalized sensitivity of neuron $i$ defined as $d'_i \equiv f'_i/\sigma_i$. **a)** Data replotted from Fig 5c of [Bondy et al. (2018)], who isolated the task-dependent component of noise correlations in macaque V1 and found a strong relation between elements of this correlation matrix and neural sensitivities ($r = 0.61$, $p < 0.001$, from original paper). This relationship between $Corr_{ij}$ and $d'_i d'_j$ follows from the relationship between $\Sigma_{ij}$ and $f'_i f'_j$ as in Eq (8). **b)** Data replotted from Fig 2d of Rabinowitz et al. (2015) [26], who found that the strength of top-down 'modulator' weights is linearly related to $d'$. **c)** Illustration of the task-dependent and arbitrary nature of $\mathbf{f'}$: each row of grating stimuli define a different discrimination task, with each task defining a different $\mathbf{f'}$ vector in neural space. For instance, relative to fine discrimination around vertical (green), one may modify the spatial frequency (blue), the phase (purple), the reference orientation to horizontal (red), or consider a coarse discrimination between vertical and horizontal (yellow). The span of possible $\mathbf{f'}$ directions is large. Hence, the probability is small of finding that the specific $\mathbf{f'}$ direction corresponding to a given task is aligned with a leading principal component of measured noise covariance, unless this alignment is the result of learning and/or performing the task.

different coarse orientation tasks. After removing the shared component of noise correlations between tasks, the *task-specific* component was found to align with $\mathbf{d'd'}^\top$ structure as predicted by Eq (8) (Fig 5A) (note that proportionality between covariance and $\mathbf{f'f'}$ is equivalent to a proportionality between correlation and $\mathbf{d'd'}$ after dividing both sides by $\sigma_i \sigma_j$). While this is encouraging for our theory, feedforward explanations of the results of [27] cannot be entirely ruled out because of the long retraining time between tasks. Cohen and Newsome [22] recorded from pairs of neurons in area MT while two monkeys switched discrimination tasks on a trial-by-trial basis and found that correlations also changed in a manner that is consistent with but not definitive of $\mathbf{f'f'}^\top$ structure (see Box 2 in [37]). By aligning the task to the recorded neurons, the results of [22] only measure a narrow slice of the full set of possible covariances (roughly $f'_i \approx f'_j$ and $f'_i \approx -f'_j$ in the two tasks). Building on these studies, a stronger test for task-dependent $\mathbf{f'f'}^\top$−covariance structure would be to interleave tasks on a trial-by-trial basis (as done by [22]) while recording from a large and random set of neurons (as done by [27]). A strong test of the dependence of structured noise covariance on feedback would be to causally manipulate cortico-cortical feedback during such a task, e.g. by cooling.

Another approach to differentiate between feedforward and feedback contributions to neural variability is to *statistically* isolate them within a single task using a sufficiently powerful regression model. [Rabinowitz et al.] [26] used this type of approach to infer the primary top-down modulators of V4 responses in a change-detection task. They found that the two most important short-term modulators were indeed aligned with the vector corresponding to neurometric sensitivity in the task (data replotted in Fig 5B). This is consistent with our predictions if the latent 'modulator' reflects variations in the subject's belief state, which in this case would be a graded belief in the two categories of "change" versus "no change." Indeed, Rabinowitz et al (2015) further reported that the modulator state correlates with subjects' decisions.

A final line of evidence comes from simultaneous recordings of large populations of neurons, and analyzing the magnitude of neural variability in the task-specific $\mathbf{f'}$ direction relative to other directions. While it is well established that there is often low-rank structure to noise covariance (hypothesized to be at least partly due to recurrent dynamics [100, 101]), and while it is expected that at least some variance is in the $\mathbf{f'}$−direction due to feedforward noise [33],

these in principle need not be aligned. Even in orientation discrimination tasks, the $\mathbf{f'}$ direction depends on arbitrary experimental choices such as the particular discriminanda (e.g. fine discrimination at vertical or horizontal, or coarse discrimination of cardinal or oblique targets) and on other arbitrary features of the stimulus such as contrast, phase, spatial frequency, etc (Fig 5C). This makes the $\mathbf{f'}$ direction largely *arbitrary*, and implies that finding high variance in the $\mathbf{f'}$ direction for a particular task, out of the many possible $\mathbf{f'}$ directions for other possible tasks, is highly unlikely under the assumption of a fixed, task-independent covariance structure. In light of this, the results of Rabinowitz et al [26], as well as other recent findings of alignment between $\mathbf{f'}$ and the low-rank modes of covariance [27–29], suggests that task information directly enters into the noise covariance structure, e.g. due to feedback of beliefs or the "filtering" of noise by a task-specific prior as we have highlighted here.

### Inferring variable internal beliefs from sensory responses

We have shown that internal beliefs about the stimulus induce corresponding structure in the correlated variability of sensory neurons' responses (Fig 6A). Conversely, this means that the statistical structure in sensory responses can be used to infer properties of those beliefs.

In order to demonstrate the usefulness of this approach, we used it to infer the structure of a ground-truth model [34] from synthetic data. The model discriminated either between a vertical and a horizontal grating (cardinal context), or between a −45deg and +45deg grating (oblique context). The model was given an unreliable (80/20) cue as to the correct context before each trial, and thus had uncertainty about the exact context. We simulated the responses of a population of primary visual cortex neurons with oriented receptive fields that perform sampling-based inference over image features. The resulting noise correlation matrix—computed for *zero-signal trials*—has a characteristic structure in qualitative agreement with empirical observations (Fig 6B) [27].

Interestingly, the resulting simulated neural responses have five significant principal components (PCs) (Fig 6C and 6D). Knowing the preferred orientation of each neuron allows us to interpret the PCs as directions of variation in the model's belief about the current orientation. For instance, the elements of the first PC (blue in Fig 6C) are largest for neurons preferring vertical and negative for those preferring horizontal orientation, indicating that there is

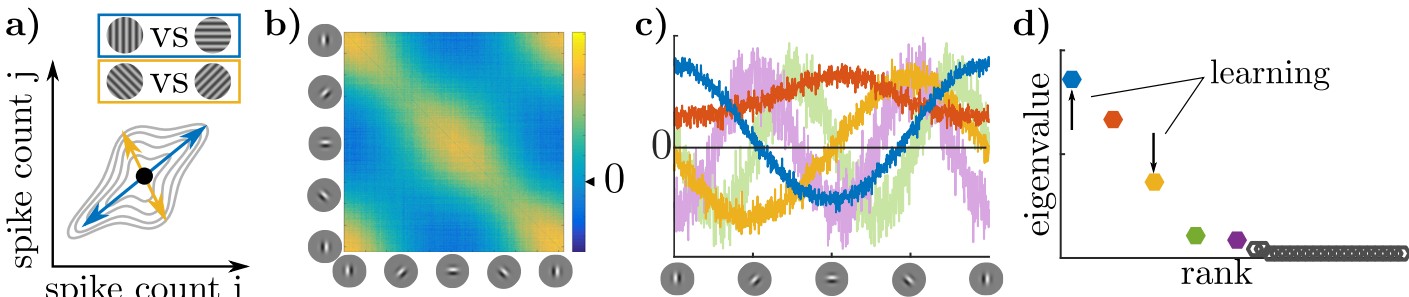

**Fig 6. Inferring the structure of internal beliefs about a task. a)** Trial-to-trial fluctuations in the posterior beliefs about **x** imply trial-to-trial variability in the mean responses representing that posterior. Each such 'belief' yields increased correlations in a different direction in **r**. The model in (b-d) has uncertainty in each trial about whether the current task is a vertical-horizontal orientation discrimination or an oblique discrimination. **b)** Correlation structure of simulated sensory responses during discrimination task. Neurons are sorted by their preferred orientation (based on [34]). **c)** Eigenvectors of correlation matrix (principal components) plotted as a function of neurons' preferred orientation. The blue vector corresponds to fluctuations in the belief that either a vertical or horizontal grating is present, and the yellow corresponds to fluctuations in the belief that an obliquely-oriented grating is present. See Methods for other colors. **d)** Corresponding eigenvalues color-coded as in (c). Our results on variable beliefs ($\pi$) predict an increase over learning in the eigenvalue corresponding to fluctuations in belief for the correct task, while our results on filtering noise predict a relative increase in the task-relevant eigenvalue compared with variance in other tasks' directions (e.g. if both blue and yellow decrease, but yellow more so).

trial-to-trial variability in the model's internal belief about whether "there is a vertical grating and not a horizontal grating"—or vice versa—in the stimulus, corresponding to the $\mathbf{f}'$–axis of the cardinal task corresponding to the most likely task in our simulations. Analogously, one can interpret the third PC (yellow in Fig 6C and 6D) as corresponding to the belief that a +45˚ grating is being presented, but not a −45˚ grating, or vice versa. This is the $\mathbf{f}'$-axis for the wrong (oblique) task context, reflecting the fact that the model maintained some uncertainty about which was the correct task in a given trial. The remaining PCs in Fig 6C and 6D correspond to task-independent variability (see S3 Fig). This example demonstrates that at least some of a subject's beliefs about the structure of the task can be read off of the principal directions of variation in neural space, possibly revealing cases of model mismatch, such as finding significant variability corresponding to the "other" task in the third principal component here. An interesting direction for future work will be to compare such an analysis to behavioral data, which can also reveal deviations between the experimenter-defined task and a subject's model of the task, for instance through the use of psychophysical kernels.

Maintaining uncertainty about the task itself is the optimal strategy from the subject's perspective given their imperfect knowledge of the world. However, when compared to perfect knowledge of context, it decreases behavioral performance which is optimal when the internal model learned by the subject matches the experimenter-defined one—an ideal which Bayesian learners approach over the course of learning. An empirical prediction, therefore, is that eigenvalues corresponding to the correct task-defined stimulus dimension will increase with learning, while eigenvalues representing other tasks should decrease. Furthermore, the shape of the task-relevant eigenvectors should be predictive of psychophysical task-strategy. Importantly, they constitute a richer, higher-dimensional, characterization of a subject's decision strategy than psychophysical kernels or CPs [105], and can leverage simultaneous recordings from neuronal populations of increasing size.

## Discussion

We derived a novel analytical link between the two dominant frameworks for modeling sensory perception: probabilistic inference and neural population coding. Under the assumption that neural responses represent posterior beliefs about latent variables, we showed how a fundamental self-consistency relationship of Bayesian inference gives rise to differential correlations in neural responses that are specific to each task and that emerge over the course of learning. We identified two mechanisms by which this can happen: (i) variable beliefs about the stimulus category that project back to the sensory population under study, and (ii) interactions between a learned prior and variable likelihoods, such that noise in the $\mathbf{f}'$ direction is less-suppressed than other directions. Our first results (i) concern optimal feedback of decision-related information in a discrimination task and its effect on both choice probabilities and noise correlations. While this result required almost no assumptions about how distributions are encoded in neural responses, it assumed negligible noise. Incorporating non-negligible noise, we identified a novel class of "Linear Distributional Codes" for which the same predictions hold. Our second set of results (ii) were obtained numerically, and we found that self-consistent inference can, in some cases, suppress irrelevant dimensions of noise and also leads to an increase in differential correlations. Re-examining existing data, we found evidence for such task-specific noise correlations aligned with neural sensitivity, which both supports the hypothesis that sensory neurons encode posterior beliefs and provides a novel explanation for previously puzzling empirical observations. Finally, we illustrated how measurements of neural responses can in principle be used to infer aspects of a subject's internal trial-by-trial beliefs in the context of a task.

## Feedback, correlations, and population information

We began by distinguishing two principal ways in which correlated neural variability arises in a Bayesian inference framework. The first is neural variability in the encoding of a fixed posterior, which has previously been studied primarily in the context of neural sampling codes [48, 52, 55, 67, 68]. The second, which we study here, is variability in the posterior itself, which is shaped by two further mechanisms. The first source of variability in the posterior itself that we study comes from variability in task-relevant categorical belief ($\pi$), projected back to sensory populations during each trial. We showed that variable categorical beliefs induce commensurate choice probabilities and neural *co*variability in approximately the $\mathbf{f}'$−direction, assuming the subject learns optimal statistical dependencies. This holds for general distributional codes if noise is negligible and stimuli are narrowly concentrated in the sub-threshold regime. Note that this latter requirement of a narrow stimulus distribution is partly an artifact of our mathematical approach, which was to show *perfect* alignment between feedback and $\mathbf{f}'$ in a limit, while making as few assumptions as possible about the brain's internal model or how distributions are encoded in neural activity. We expect our results to degrade gracefully, e.g. *near-perfect* alignment between decision-related feedback and $\mathbf{f}'$ may be predicted given only mild restrictions on distributional codes and more permissive assumptions about the experimental distribution of stimuli (see S1 Text for further discussion). When noise is non-negligible, the same alignment result additionally holds for a newly-identified class of Linear Distributional Codes (LDCs). A second source of variation in the posterior itself is task-independent noise that interacts with a task-dependent prior. Although not solved analytically, we found in simulation that the task-dependent component of this variability increased differential correlations after learning in three out of four simulated conditions, through the mechanism of suppressing variability that is inconsistent with the task-specific prior. The first set of these simulations highlight the fact that empirically detecting effects of a changing prior requires using stimuli that are "open to interpretation," i.e. stimuli for which $p_b(\mathbf{E}|\mathbf{x})$ is relatively wide [102]. The second set of these simulations showed that an increase in differential correlations is not guaranteed by suppressing noise that is inconsistent with the prior, but depends on the particular relationships between stimuli, noise, and the brain's internal model. While this limits the power of this noise-suppression mechanism to *predict* differential correlations that increase with learning, such a mechanism may nonetheless play an important role in explaining existing data. We hope that future work on this noise-suppression mechanism will lead to new empirically-decidable signatures of particular kinds of internal models.

Both of these two sources of task-dependent posterior variability depend on feedback: in one case there is dynamic feedback of a particular belief $\pi$ each trial, and in the other case there is task-dependent (but belief-independent) feedback that sets a static prior each trial, then interacts with noise in the likelihood, analogous to models of "state-dependent" recurrent dynamics [100, 101, 106].

Our results directly address several debates in the field on the nature of feedback to sensory populations. First, they provide a function for the apparent 'contamination' of sensory responses by choice-related signals [17, 26, 27, 31, 32, 44]: top-down signals communicate task-relevant expectations, not reflecting a choice *per se* but integrating information in a statistically optimal fashion as previously proposed [64, 86]. Second, if this feedback is variable across trials, reflecting the subject's variable beliefs, it induces choice probabilities that are the result of both feedforward and feedback components [31, 32, 34, 107].

Our results suggest that at least some of measured "differential" covariance may be due to optimal feedback from internal belief states, or as the interaction between task-independent noise and a task-specific prior. In neither case is information necessarily more limited as the

result of learning, despite an increase in so-called "information limiting" correlations. In the first case, while feedback of belief ($\pi$) biases the sensory population, that bias might be accounted for by downstream areas [21, 30]. In principle, these variable belief states could *add* information to the sensory representation if they are *true* [37] which is not the case in psychophysical tasks with uncorrelated trials that deviate from the temporal statistics of the natural world [87]. In the second case, the noise in the $\mathbf{f}'$−direction *does* limit information, but to the same extent as before learning; there is not necessarily *further* reduction of information by "shaping" the noise with a task-specific prior. For a fixed population size, it is covariance in the $\mathbf{f}'$−direction, not correlation, that ultimately affects information. Both of these possibilities call for caution when interpreting studies that estimate the information content of sensory populations by estimating the amount of variance (or correlation) in the $\mathbf{f}'$− (or $d'$−) direction. These insights must be taken into account when interpreting recent experimental evidence for strong differential correlations [28, 29]. Take the recent results of [29], for example, who analyzed V1 calcium imaging data in mouse V1 and found that the $\mathbf{f}'$ direction was among the top 5 principal components among thousands of neurons. We emphasize that these results are highly statistically surprising from the feedforward perspective when considering the arbitrariness of the $\mathbf{f}'$ direction, which corresponds to the specific task in a given experiment (Fig 5C).

## Interpreting low-dimensional structure in population responses

Much research has gone into inferring latent variables that contribute to low-dimensional structure in the responses of neural populations [108–110]. Our results suggest that at least some of these latent variables can usefully be characterized as internal beliefs about sensory variables. We showed in simulation that the influence of each latent variable on recorded sensory neurons can be interpreted in the stimulus space using knowledge of the stimulus-dependence of each neuron's tuning function (Fig 6). Our results are complementary to *behavioral* methods to infer the shape of a subject's prior [111], but have the advantage that the amount of information that can be collected in neurophysiology experiments far exceeds that in psychophysical studies allowing for richer characterization of the subject's internal model [112].

## A fresh look at distributional coding

We introduced a general notation for distributional codes, $\mathcal{R}$, that encompasses nearly all previously proposed distributional codes. Thinking of distributional codes in this way—as a map from an implicit space $p_b(\mathbf{x})$ to observable neural responses $p(\mathbf{r})$—is reminiscent of early work on distributional codes [45], and emphasizes the convenience of computation, manipulation, and decoding of $p_b(\mathbf{x}|...)$ from $\mathbf{r}$ rather than its spatial or temporal allocation of information *per se* [3, 5, 47]. Our results leverage this generality and show that there are properties of Bayesian computations that are identifiable in neural responses without strong commitments to their algorithmic implementation. Rather than assuming an approximate inference algorithm (e.g. sampling) then deriving predictions for neural data, future work might productively work in the reverse direction, asking what class of generative models ($\mathbf{x}$) and encodings ($\mathcal{R}$) are consistent with some data. As an example of this approach, we observe that the results of Berkes et al. (2011) are consistent with any LDC, since LDCs have the property that the average of encoded distributions equals the encoding of the average distribution, as their empirical results suggest [4].

Distinguishing between linear and nonlinear distributional codes is complementary to the much-debated distinction between parametric and sampling-based codes. LDCs include both

sampling codes where samples are linearly related to firing rate [34, 48–51, 91], parametric codes where firing rates are proportional to expected statistics of the distribution [7, 45, 56, 99]. Examples of distributional codes that are *not* LDCs include sampling codes with nonlinear embeddings of the samples in **r** [52, 53, 55, 69], parametric codes in which the *natural parameters* of an exponential family are encoded [58, 60, 62, 80], as well as "expectile codes" recently proposed based on ideas from reinforcement learning [113, 114].

Our focus on firing rates and spike count covariance is motivated by connections to rate-based encoding and decoding theory. We do not assume that they are the sole carrier of information about the underlying posterior $p_b(\mathbf{x}|\ldots)$, but simply statistics of a larger spatio-temporal space of neural activity, **r** [85, 93]. For many distributional codes, firing rates are only a summary statistic, but they nonetheless provide a window into the underlying distributional representation.

## Posterior versus likelihood distributional coding

Probabilistic Population Codes (PPCs) have been instrumental for the field's understanding of the neural basis of inference in perceptual decision-making. However, they are typically studied in a purely feedforward setting assuming a representation of the likelihood, not posterior [58, 60]. In contrast, Tajima et al. (2016) modeled a PPC encoding the posterior and found that categorical priors bias neural responses in the $\mathbf{f}'$−direction, consistent with our results [70]. This clarifies and formalizes a connection between Tajima et al (2016) and Haefner et al (2016) (who simulated sampling-based hierarchical inference and found excess variance in the $\mathbf{f}'$−direction): the crucial ingredient in both is the feedback of categorical beliefs rather than the choice of sampling or parametric representation *per se*.

The assumption that sensory responses represent posterior beliefs through a general encoding scheme agrees with empirical findings about the top-down influence of experience and beliefs on sensory responses [64, 107, 115]. It also relates to a large literature on association learning and visual imagery (reviewed in [116]). In particular, the idea of "perceptual equivalence" [117] reflects our starting point that changes in the posterior belief (and hence changes in the percept) can be the result of either changes in sensory inputs or changes in prior expectations. When prior expectations vary, they manifest as correlated neural variability which can be understood in terms of equivalent changes in sensory inputs (Fig 6). Through learning, expectations come to align with past variations in stimuli (Fig 3) leading to testable predictions for the induced structure in neural covariability (Eq (8)).

## Attention and learning signals

Our results imply that the increase in alignment with neural sensitivity, $\mathbf{f}'$, for both differential correlations and choice probabilities, depends on the extent to which inference in the brain is *approximate*. Attention, which is classically characterized as allocating limited resources [118], can be expected to improve the approximation. This could happen either by reducing excess variability in expectations about the stimulus category, or by reducing noise in the likelihood. This would be compatible with empirical data which show that both noise correlations [40–42] and choice probabilities [31] are reduced in high-attention conditions.

Also complementary to our work is a mechanistic network model by [119] in which choice-related feedback was shown to be helpful for the learning of a categorical representation in areas intermediate to sensory areas and decision-making areas. As in our work, correlated variability and decision-related signals emerge as the result of task-specific changes in feedback connections.

[44] provided an alternative rationale for the brain to introduce additional differential correlations by showing that they make it easier for a decision-making area to learn the linear decoding weights to extract task-relevant information from the sensory responses.

### Limitations and deviations from assumptions

Our derivations implicitly assumed that the feedforward encoding of sensory information, i.e. the likelihood $p_b(\mathbf{E}|\mathbf{x})$, remains unchanged between the compared conditions. This is well-justified for lower sensory areas in adult subjects [120], or when task contexts are switched on a trial-by-trial basis [22]. However, it is not necessarily true for higher cortices [121], especially when the conditions being compared are separated by extended periods of task (re)training [27]. In those cases, changing sensory statistics may lead to changes in the feedforward encoding, and hence the nature of the represented variable $\mathbf{x}$ [122, 123].

In the context of our theory, there are three possible deviations from our assumptions that can account for empirical results of a less-than-perfect alignment with $\mathbf{f}'$ [42, 124]—each of them empirically testable. First, it is plausible that only a subset of sensory neurons represent the posterior, while others represent information about necessary ingredients (likelihood, prior), or carry out other auxiliary functions [50, 53]. Our predictions are most likely to hold among layer 2/3 pyramidal cells, which are generally thought to encode the *output* of cortical computation in a given area, i.e. the posterior in our framework [125]. Second, subjects may not learn the task *exactly* implying a difference between the experimenter-defined task and the subject's "subjective" $\mathbf{f}'$–direction for which our predictions apply. This explanation could be verified using psychophysical reverse correlation to identify the task the subject has effectively learned from behavioral data, in which case we would predict alignment between decision-related feedback and the subject's internal notion of the discriminanda. However, not all deviations between the subject's internal model and the experimenter model result in deviations from our predictions. For instance, the subject's categorical decision may be mediated by other variables between $\mathbf{x}$ and $C$, including an internal estimate of $s$ itself. In fact, one can treat a scalar internal estimate of $s$ much like a large set of fine-grained *internal* categorical distinctions beyond the two categories imposed by the experiment, such as distinguishing "large positive $s$" from "small positive $s$." Trial-by-trial variations in such fine-grained categorical expectations should, unsurprisingly, feed back and modulate sensory neurons in the $\mathbf{f}'$ direction, even if the categories defined by the experimenter are broad (discussed further in S1 Text). Third, some misalignment between $\mathbf{f}'$ and decision-related feedback may be indicative of significant task-independent noise in the presence of a nonlinear distributional code, which could be tested by manipulating the amount of external noise in the stimulus. If decision-related feedback is found to align with $\mathbf{f}'$ under a range of noise levels and tasks, this would be evidence in favor of a Linear Distributional Code. Given the generality of our derivations, and subject to the caveats just discussed, empirically finding misalignment between $\mathbf{f}'$ and top-down decision-related signals may be interpreted as evidence that neurons do not encode posterior distributions, that the encoding is instead nonlinear (i.e. not an LDC), or that task-learning is suboptimal. In the latter two cases, we would nonetheless predict low-rank task-dependent changes in neural covariability, albeit not aligned with $\mathbf{f}'$.

## Methods

### Self-consistency is implied by learning the data distribution

Following previous work [3, 64, 72, 74], we assume that the brain has learned an implicit hierarchical generative model of its sensory inputs, $p_b(\mathbf{E}|\mathbf{x})$, in which perception corresponds to

inference of latent variables, $\mathbf{x}$, conditioned on those inputs. The subscripted distributions $p_b(\cdot)$ and $p_e(\cdot)$ refer to the brain's internal model and the experimenter's "ground truth" model, respectively.

Here we provide a brief proof that, once a probabilistic model learns the data distribution, its average posterior is equal to its prior [3, 4, 93]. This proof is not novel, and a similar argument can be found in the supplement of Berkes et al (2011). Assume that $p_b(\mathbf{E}) = p_e(\mathbf{E})$. Then,

$$
\begin{aligned}
\int_{\mathbf{E}} p_b(\mathbf{E}, \mathbf{x})d\mathbf{E} = \int_{\mathbf{E}} p_b(\mathbf{x}|\mathbf{E})p_b(\mathbf{E})d\mathbf{E} &= p_b(\mathbf{x}) \\
&= \mathbb{E}_{p_e(\mathbf{E})}[p_b(\mathbf{x}|\mathbf{E})]
\end{aligned}
, \tag{11}
$$

where the top line (the prior) follows from the definition of marginalization, and the bottom line (the average posterior) follows from the assumption. In other words, the self-consistency rule we use—that the prior equals the average posterior—is implied by the brain's internal model learning the statistics of the task. Further discussion and an alternative approach to this proof is given in S1 Text.

## Optimal self-consistent expectations over x

In the classic two-alternative forced-choice paradigm, the experimenter parameterizes the stimulus with a scalar variable $s$ and defines category boundary which we will arbitrarily denote $s = 0$. If there is no external noise, the scalar $s$ is mapped to stimuli by some function $\mathbf{E}(s)$, for instance by rendering grating images at a particular orientation. In the case of noise, below, we consider more general stimulus distributions $p_e(\mathbf{E}|s)$.

We assume that the brain does not have an explicit representation of $s$ but must form an internal estimate of the category each trial, $C$, based on the variables represented by sensory areas, $\mathbf{x}$ [91]. From the "ground truth" model perspective, stimuli directly elicit perceptual inferences—this is why we include $p_e(\mathbf{x}|\mathbf{E})$ as part of the experimenter's model. In the brain's internal model, on the other hand, the stimulus is assumed to have been generated by causes $\mathbf{x}$, which are, in turn, *jointly* related to $C$. These models imply the following conditional independence relations:

$$
\begin{aligned}
p_e(C, s, \mathbf{E}, \mathbf{x}) &= p_e(C)p_e(s|C)p_e(\mathbf{x}|\mathbf{E})\delta(\mathbf{E} - \mathbf{E}(s)) \\
&= p_e(C)p_e(s|C)p_e(\mathbf{x}|\mathbf{E}(s)) \\
p_b(\mathbf{E}, \mathbf{x}, C) &= p_b(C)p_b(\mathbf{x}|C)p_b(\mathbf{E}|\mathbf{x}) .
\end{aligned}
$$

We assume the brain learns the joint distribution $p_b(\mathbf{x}, C)$ that maximizes reward, or equivalently that best matches the ground-truth distribution $p_e(C, \mathbf{x})$ in expectation. This entails a conditional distribution "decoding" $C$ from $\mathbf{x}$ of the form

$$
p_b(C|\mathbf{x}) = \int_s p_e(C|s)p_e(\mathbf{E}(s)|\mathbf{x})ds . \tag{12}
$$

We next derive the reciprocal influence of $C$ on $\mathbf{x}$ (Eq (2) in the main text) by applying Bayes' rule to Eq (12):

$$
\begin{aligned}
\mathrm{p_b}(\mathbf{x}|C) &= \frac{\mathrm{p_b}(\mathbf{x})}{\mathrm{p_b}(C)} \int_s \mathrm{p_e}(C|s)\mathrm{p_e}(\mathbf{E}(s)|\mathbf{x})\mathrm{d}s \\
&= \frac{\mathrm{p_e}(C)}{\mathrm{p_b}(C)} \int_s \mathrm{p_e}(s|C)\mathrm{p_e}(\mathbf{x}|\mathbf{E}(s))\mathrm{d}s \\
&= \int_s \mathrm{p_e}(s|C)\mathrm{p_b}(\mathbf{x}|\mathbf{E}(s))\mathrm{d}s \\
\mathrm{p_b}(\mathbf{x}|C) &= \mathbb{E}_{\mathrm{p_e}(s|C)}[\mathrm{p_b}(\mathbf{x}|\mathbf{E}(s))] \qquad\qquad \text{((2) restated)}
\end{aligned}
$$

The substitution of $\mathrm{p_b}$ for $\mathrm{p_e}$ in the third line follows from the fact that, even from the perspective of an external observer, $\mathrm{p_e}(\mathbf{x}|s)$ is the inference made *by the brain* about $\mathbf{x}$ induced by the stimulus $\mathbf{E}(s)$. Hence, $\mathrm{p_e}(\mathbf{x}|s)$ is equivalent to $\mathrm{p_b}(\mathbf{x}|\mathbf{E}(s))$. The fractions $\mathrm{p_e}(C)/\mathrm{p_b}(C)$ and $\mathrm{p_b}(\mathbf{x})/\mathrm{p_e}(\mathbf{x})$ become one, assuming that the subject learns the correct categorical prior on $C$ and a consistent internal model. We note that this distribution can be learned even if $s$ is not directly observable by the brain, since its model has access to the true category labels if subjects are informed of the correct answer each trial, as well as to each individual posterior $\mathrm{p_b}(\mathbf{x}|s)$, as this is what we assume is represented by the sensory area.

As described in the main text we marginalize over the subject's belief in the category, $\pi = \mathrm{p_b}(C = 1)$, to get an expression for expectations on $\mathbf{x}$ given the belief (Eq (3)). Unlike $C$, $\pi$ is not a random variable in the generative model but the *parameter* defining the subject's belief about the binary variable $C$. The resulting posterior on $\mathbf{x}$, abbreviated in Eq (4), is given by

$$
\begin{aligned}
\mathrm{p_b}(\mathbf{x}|\mathbf{E}(s), \pi) &= \frac{\mathrm{p_b}(\mathbf{E}(s)|\mathbf{x})\mathrm{p_b}(\mathbf{x}|\pi)}{\mathrm{p_b}(\mathbf{E}(s)|\pi)} \qquad\qquad \text{((4) restated)} \\
&= \mathrm{p_b}(\mathbf{E}(s)|\mathbf{x})\left[\frac{\pi\mathrm{p_b}(\mathbf{x}|C=1) + (1-\pi)\mathrm{p_b}(\mathbf{x}|C=2)}{\pi\mathrm{p_b}(\mathbf{E}(s)|C=1) + (1-\pi)\mathrm{p_b}(\mathbf{E}(s)|C=2)}\right],
\end{aligned}
\tag{13}
$$

We assume that the category boundary $s = 0$ is itself equally likely to occur conditioned on each category (usually true by definition), but note that this is *not* a requirement that the categories are *a priori* equally likely. This simplifies Eq (13) when conditioning on $s = 0$:

$$
\mathrm{p_b}(\mathbf{x}|\mathbf{E}(s=0), \pi) = \frac{\mathrm{p_b}(\mathbf{E}(s=0)|\mathbf{x})}{\mathrm{p_b}(\mathbf{E}(s=0))}\left[\pi\mathrm{p_b}(\mathbf{x}|C=1) + (1-\pi)\mathrm{p_b}(\mathbf{x}|C=2)\right]. \tag{14}
$$

**Proof of (5): Symmetry between changes in the stimulus $s$ and categorical belief $\pi$.**   Our first main result is the approximate proportionality in (5), restated here:

$$
\left.\frac{\mathrm{d}}{\mathrm{d}s}\mathrm{p_b}(\mathbf{x}|\mathbf{E}(s), \pi = {}^1\!/_2)\right|_{s=0} \stackrel{\sim}{\propto} \left.\frac{\mathrm{d}}{\mathrm{d}\pi}\mathrm{p_b}(\mathbf{x}|\pi, \mathbf{E}(s=0))\right|_{\pi={}^1\!/_2}. \tag{(5) restated}
$$

We use $\pi = {}^1\!/_2$ to denote the true prior over categories, $\mathrm{p_e}(C)$. This is often 50/50, but our results generalize to cases with asymmetric prior probabilities, in which case derivatives must be taken around the point $s = 0$, $\pi = \mathrm{p_e}(C)$.

Since $s = 0$ is fixed in the right-hand-side of (5), the total derivative with respect to $\pi$ equals its partial derivative, assuming that there are no *additional* internal variables that are dependent on both $\mathbf{x}$ and $\pi$. In the left-hand-side of (5), the total derivative with respect to $s$ includes two terms, one due to the direct effect of $s$ on the posterior, and the other due to the mean

 

dependence of $\pi$ on $s$, since changes in $s$ elicit changes in the subject's beliefs:

$$\frac{\mathrm{d}}{\mathrm{d}s}\mathrm{p}_\mathrm{b}(\mathbf{x}|\mathbf{E}(s))\bigg|_{s=0} = \frac{\partial}{\partial s}\mathrm{p}_\mathrm{b}(\mathbf{x}|\mathbf{E}(s), \pi = {}^1\!/_2)\bigg|_{s=0} + \frac{\partial\pi}{\partial s}\frac{\partial}{\partial\pi}\mathrm{p}_\mathrm{b}(\mathbf{x}|\mathbf{E}(s=0), \pi)\bigg|_{\pi={}^1\!/_2}.$$

Below, we will replace $\mathrm{p}_\mathrm{b}\big(\mathbf{x}|\mathbf{E}(s), \pi = \frac{1}{2}\big)$ with $\mathrm{p}_\mathrm{b}(\mathbf{x}|\mathbf{E}(s))$ to reduce notational clutter since $\pi = {}^1\!/_2$ corresponds to marginalizing over categories with the true prior. The second partial derivative term in the previous equation is equal to the right-hand-side of (5), scaled by $\partial\pi/\partial s$, and hence does not affect the overall proportionality in (5). To prove the approximate proportionality in (5), we therefore need only prove proportionality in the partial derivatives:

$$\frac{\partial}{\partial s}\mathrm{p}_\mathrm{b}(\mathbf{x}|\mathbf{E}(s))\bigg|_{s=0} \tilde{\propto} \frac{\partial}{\partial\pi}\mathrm{p}_\mathrm{b}(\mathbf{x}|\pi, \mathbf{E}(s=0))\bigg|_{\pi={}^1\!/_2}. \tag{15}$$

Using a small $\Delta s$ finite-difference approximation, we rewrite t the left-hand-side of (15) as

$$\frac{\partial}{\partial s}\mathrm{p}_\mathrm{b}(\mathbf{x}|\mathbf{E}(s))\bigg|_{s=0} \approx \frac{1}{2\Delta s}\big[\mathrm{p}_\mathrm{b}(\mathbf{x}|\mathbf{E}(s=+\Delta s)) - \mathrm{p}_\mathrm{b}(\mathbf{x}|\mathbf{E}(s=-\Delta s))\big]. \tag{16}$$

While this is an approximation to the "true" derivative, it is usually a good one based on theoretical reasons (range of $s$ small in the threshold regime of psychophysical tasks) and empirical observations [27].

Next, consider the right-hand-side of (15) using the expression for the posterior conditioned on $s = 0$ (Eq (14)). The partial derivative of this posterior with respect to the belief $\pi$ is

$$\frac{\partial}{\partial\pi}\mathrm{p}_\mathrm{b}(\mathbf{x}|\pi, \mathbf{E}(s=0)) = \frac{\mathrm{p}_\mathrm{b}(\mathbf{E}(s=0)|\mathbf{x})}{\mathrm{p}_\mathrm{b}(\mathbf{E}(s=0))}\big[\mathrm{p}_\mathrm{b}(\mathbf{x}|C=1) - \mathrm{p}_\mathrm{b}(\mathbf{x}|C=2)\big].$$

Applying the self-consistency constraint implied by learning (i.e. substituting in Eq (2) to the terms inside the brackets), this becomes

$$\frac{\partial}{\partial\pi}\mathrm{p}_\mathrm{b}(\mathbf{x}|\pi, \mathbf{E}(s=0)) = \frac{\mathrm{p}_\mathrm{b}(\mathbf{E}(s=0)|\mathbf{x})}{\mathrm{p}_\mathrm{b}(\mathbf{E}(s=0))}\Big[\mathbb{E}_{\mathrm{p}_\mathrm{e}(s|C=1)}[\mathrm{p}_\mathrm{b}(\mathbf{x}|\mathbf{E}(s))] - \mathbb{E}_{\mathrm{p}_\mathrm{e}(s|C=2)}[\mathrm{p}_\mathrm{b}(\mathbf{x}|\mathbf{E}(s))]\Big].$$

Re-arranging terms, we arrive at

$$\frac{\partial}{\partial\pi}\mathrm{p}_\mathrm{b}(\mathbf{x}|\pi, s=0) = \frac{\mathrm{p}_\mathrm{b}(\mathbf{x}|\mathbf{E}(s=0))}{\mathbb{E}_{\mathrm{p}_\mathrm{e}(s)}[\mathrm{p}_\mathrm{b}(\mathbf{x}|\mathbf{E}(s))]}\Big[\mathbb{E}_{\mathrm{p}_\mathrm{e}(s|C=1)}[\mathrm{p}_\mathrm{b}(\mathbf{x}|\mathbf{E}(s))] - \mathbb{E}_{\mathrm{p}_\mathrm{e}(s|C=2)}[\mathrm{p}_\mathrm{b}(\mathbf{x}|\mathbf{E}(s))]\Big], \tag{17}$$

where we have used the identity $\mathrm{p}_\mathrm{b}(\mathbf{x}) = \mathbb{E}_{\mathrm{p}_\mathrm{e}(s)}[\mathrm{p}_\mathrm{b}(\mathbf{x}|\mathbf{E}(s))]$ to write the denominator of the fraction outside the brackets as expectations over $s$. This identity is valid because we assumed subjects have completely learned the task, so the *self-consistency* rule holds that the prior $\mathrm{p}_\mathrm{b}(\mathbf{x})$ equals the average posterior seen in the task [3, 4, 93].

Having re-arranged terms, we must now establish conditions under which (16) and (17) are proportional. While they appear similar by inspection, they are not proportional in general because so far we have placed no restrictions on the experimenter's distribution of stimuli $\mathrm{p}_\mathrm{e}(s)$. We therefore next consider the special case of sub-threshold tasks. One way to formalize this mathematically is by taking the limit of (17) as $\mathrm{p}_\mathrm{e}(s)$ approaches a Dirac delta around $s = 0$, as this appears to result in agreement between the individual terms of (17) and (16). However, in this limit (17) itself goes to zero (indeed, it should be expected that beliefs are irrelevant in a task that has zero variation in stimuli).

This suggests an approximate solution by breaking the problem into two limiting processes: one in which the distribution of stimuli within each category concentrates on some $\pm\Delta s$, and a

 

second in which $\Delta s$ gets small (but does not reach zero). S1 Fig visualizes these two steps. To realize the first limit, we set

$$p_e(s|C=2) = (1-p_0)\delta(s-\Delta s) + p_0\delta(s-0), \tag{18}$$

and likewise for $C = 1$ and $-\Delta s$. We include the $\delta(s-0)$ term to ensure that zero-signal stimuli are always included with probability $p_0$, otherwise evaluating (17) at $s = 0$ would not be possible in practice. Marginalizing over categories, the full distribution of stimuli becomes

$$p_e(s) = \frac{(1-p_0)}{2}[\delta(s-\Delta s) + \delta(s+\Delta s)] + p_0\delta(s-0). \tag{19}$$

Substituting Eqs (18) and (19) into (17) simplifies the expectations. First, the terms inside the brackets in (17) go to

$$[\mathbb{E}_{p_e(s|C=1)}[p_b(\mathbf{x}|\mathbf{E}(s))] - \mathbb{E}_{p_e(s|C=2)}[p_b(\mathbf{x}|\mathbf{E}(s))]]$$
$$= (1-p_0)[p_b(\mathbf{x}|\mathbf{E}(s=-\Delta s)) - p_b(\mathbf{x}|\mathbf{E}(s=+\Delta s))],$$

which matches the corresponding term in (16) to the extent that $\Delta s$ is small enough to approximate the derivative $\frac{df}{ds}$. Thus, the extent to which (17) is proportional to (16) depends only on the extent to which the first term in the right-hand-side of (17) is constant, or equivalently whether $p_b(\mathbf{x}|\mathbf{E}(s=0))$ approximately equals $\mathbb{E}_{p_e(s)}[p_b(\mathbf{x}|\mathbf{E}(s))]$. Considering the special case of stimulus distributions given in (18) and (19), this near-equality condition holds as the probability of true zero-signal stimuli ($p_0$) grows, or as the category differences ($\Delta s$) shrink: an approximation to sub-threshold psychophysics conditions.

Taken together, this establishes the approximate proportionality in (15), which in turn concludes the proof of (5), in the special case of sub-threshold psychophysics. See S1 Text for further discussion of the applicability and interpretation of these limits, as well as potential relaxations.

### Encoding the posterior in neural responses

Our above derivations considered perturbations of an approximate Bayesian observer's posterior over their internal variables, $p_b(\mathbf{x}|\mathbf{E}(s), \pi)$. We next link these computational-level changes in the posterior to predictions for observable changes in neural firing rate. "Posterior coding" hypothesizes that the (possibly high-dimensional) posterior $p_b(\mathbf{x}|\mathbf{E}(s), \pi)$ is encoded in the spiking pattern of a population of neurons over some time window. We do not restrict the space of neural responses $\mathbf{r}$ to total spike counts or average spike rates, but instead consider $\mathbf{r}$ on a single trial to live in a high-dimensional "spatiotemporal" space, i.e. an $N \times B$ array of spike counts for all $N$ neurons in a population resolved into $B$ fine-timescale bins [93]. That is, $\mathbf{r} \in \mathbb{R}^{N \times B}$, where $\mathbf{r}_{ib}$ is the spike count of neuron $i$ at time $b$. This definition subsumes both "spatial" and "temporal" codes, a distinction that lies at the center of some debates over the neural representation of distributions [3, 5, 47].

We define distributional codes of the *posterior* as any encoding scheme $\mathcal{R}$ where the posterior distribution on $\mathbf{x}$ is sufficient to determine the neural response distribution over the range of possible stimuli. (Note that this excludes the possibility of separately encoding the likelihood and the prior.) Formally, we say

$$p(\mathbf{r}|s, \pi) = \mathcal{R}[p_b(\mathbf{x}|\mathbf{E}(s), \pi)](\mathbf{r}), \tag{20}$$

where $\mathcal{R}$ is a higher-order function that maps from distributions over $\mathbf{x}$ to distributions over $\mathbf{r}$. In the notation of Zemel et al (1998), the distribution on $\mathbf{r}$ is conditioned directly on the

encoded distribution on $\mathbf{x}$, or $p(\mathbf{r}|p_b(\mathbf{x}|\ldots))$. Our only restrictions on $\mathbf{x}$ and $\mathcal{R}$ are that $p_b(\mathbf{x}|\ldots)$ must change sufficiently smoothly with $s$, and $\mathcal{R}$ must be sufficiently smooth over the relevant range of stimulus values, so that the derivatives and linear approximations throughout are valid. A second restriction on $\mathbf{x}$ and $\mathcal{R}$ is that the dominant effect of $s$ on $\mathbf{r}$ must be in the mean firing rates rather than their higher-order moments of $\mathbf{r}$. While this is a theoretically complex condition to meet involving interactions between $s$, $\mathbf{x}$, and $\mathcal{R}$, it is easily verified empirically in a given experimental context: if changes to $s$ primarily influence the mean spike count, it is irrelevant whether these changes coded for the mean, variance, or higher-order moments of $p_b(\mathbf{x}|\ldots)$. If the space of $\mathbf{r}$ is the full "spatiotemporal" space of neural activity patterns, this definition encompasses all previously proposed parametric [7, 62, 70, 80], and sampling-based [34, 48, 49, 51–53] encoding schemes as special cases, among others. However, it excludes sub-populations of neurons in which only the likelihood or prior, but not the posterior, is encoded [58, 60, 126].

## Tuning curves as statistics of encoded distributions

The total spike count of neuron $i$ in terms of $\mathbf{r}$ is a function of $\mathbf{r}$ that sums responses over time bins:

$$\text{spike count}_i \equiv S_i(\mathbf{r}) = \sum_{b=1}^{B} \mathbf{r}_{ib} \,.$$

In an encoding model defined as in Eq (20), each neuron's tuning curve is thus defined by the expectation of $S_i$ at each value of the stimulus $s$:

$$f_i(s) = \mathbb{E}_{\mathbf{r} \sim \mathcal{R}[p_b(\mathbf{x}|\mathbf{E}(s))]}[S_i(\mathbf{r})] \,. \tag{21}$$

The *slope* of this tuning curve, $\frac{\mathrm{d}f_i}{\mathrm{d}s}$, is given by the chain rule:

$$\frac{\mathrm{d}f_i}{\mathrm{d}s} = \left\langle \frac{\mathrm{d}f_i}{p_b(\mathbf{x}|\mathbf{E}(s))}, \frac{\mathrm{d}p_b(\mathbf{x}|\mathbf{E}(s))}{\mathrm{d}s} \right\rangle, \tag{(1) restated}$$

where the inner product is taken between two functions, since derivatives were taken with respect to the distribution $p_b(\mathbf{x}|\mathbf{E}(s), \pi)$. Eq (1) shows how we use smoothness and linearization assumptions to decouple our analysis of changes in posteriors (e.g. $\mathrm{d}p_b/\mathrm{d}s$) from their effect on mean firing rates under arbitrary distributional encodings (e.g. $\mathrm{d}f_i/\mathrm{d}p_b$). The proportionality between $\mathrm{d}p_b/\mathrm{d}s$ due to changing stimuli and $\mathrm{d}p_b/\mathrm{d}\pi$ due to feedback of beliefs (Eq (5)) implies an analogous proportionality in neural responses:

$$\left. \frac{\mathrm{d}\mathbf{f}}{\mathrm{d}\pi} \right|_{\substack{s=0 \\ \pi = 1/2}} \widetilde{\propto} \left. \frac{\mathrm{d}\mathbf{f}}{\mathrm{d}s} \right|_{\substack{s=0 \\ \pi = 1/2}} \,. \tag{(6) restated}$$

## Derivation of (9): Feedback of categorical belief implies choice probability proportional to $d'$

We assume the subject's choice is based on their posterior belief in the stimulus category, i.e. value of $\pi$. Conditioning neural responses on choice is then equivalent to conditioning on the sign of $\pi - 1/2$ (if there is an additional stage of randomness between belief $\pi$ and behavioral choice, what follows will remain true up to a proportionality, [21]).

Let $\text{CTA}_i$ be the "choice triggered average" of neuron $i$, defined as the difference in mean response to choice 1 and choice 2. To isolate top-down effects, consider the noiseless case

where neural responses depend exclusively on $s$ (which is fixed) and $\pi$ (which is varying). We then write CTA as the difference in expected neural response between the $\pi > {}^1/_2$ and $\pi < {}^1/_2$ cases:

$$\mathrm{CTA}_i \equiv \mathbb{E}_{\pi > {}^1/_2}[f_i(s = 0, \pi)] - \mathbb{E}_{\pi < {}^1/_2}[f_i(s = 0, \pi)].$$

For small variability in $\pi$, this can be approximated linearly:

$$\mathrm{CTA}_i \approx \left( f_i(s = 0, \pi = {}^1/_2) + \Delta\pi \frac{\mathrm{d}f_i}{\mathrm{d}\pi} \right) - \left( f_i(s = 0, \pi = {}^1/_2) - \Delta\pi \frac{\mathrm{d}f_i}{\mathrm{d}\pi} \right)$$

$$= 2\Delta\pi \frac{\mathrm{d}f_i}{\mathrm{d}\pi}.$$

Substituting in the proportionality $\mathrm{d}\mathbf{f}/\mathrm{d}\pi \mathrel{\tilde{\propto}} \mathrm{d}\mathbf{f}/\mathrm{d}s$ (6), it follows that $\mathrm{CTA}_i \mathrel{\tilde{\propto}} f_i'$. Dividing both sides of this proportionality by the standard deviation of the neuron's response, $\sigma_i$, and incorporatig the fact that $\mathrm{CP}_i - \frac{1}{2} \propto \mathrm{CTA}_i/\sigma_i$ [19, 20], we arrive at the following equation for the *top-down* component of choice probability after learning:

$$\mathrm{CP}_i - \frac{1}{2} \propto f_i'/\sigma_i \equiv d_i', \qquad\qquad ((9)\text{restated})$$

where $d'$ is the "d-prime" sensitivity measure from signal detection theory [95].

## Derivation of (8): Feedback of categorical belief implies differential covariance

Consider any scalar variable $a$ that linearly shifts neural responses in an arbitrary direction $\mathbf{u}$, above and beyond all of the other factors influencing the population (denoted "..."):

$$\mathbf{f}(\dots, a) = \mathbf{f}(\dots) + a\mathbf{u} + \text{noise}.$$

When $a$ varies from trial to trial (independently of other factors in "..."), it adds a rank-1 component to the covariance matrix:

$$\Sigma = \Sigma^{\text{intrinsic}} + \mathrm{var}(a)\mathbf{u}\mathbf{u}^\top,$$

where $\Sigma^{\text{intrinsic}}$ is the covariance due to all other factors, i.e. due to neural noise and variability in any of the terms in "..." [37].

It follows that *variability* in the posterior along $\mathrm{d}p_b/\mathrm{d}s$ manifest as covariability among neurons in the $\mathbf{f}'\mathbf{f}'^\top$ direction. The noise covariance structure due to $\mathrm{var}(\pi)$ is predicted to be

$$\Sigma \approx \Sigma^{\text{intrinsic}} + \underbrace{\alpha^2\mathrm{var}(\pi)\mathbf{f}'\mathbf{f}'^\top}_{\Sigma^{\text{belief}}}. \qquad\qquad (22)$$

$\Sigma^{\text{intrinsic}}$ may be thought of as neural noise above and beyond variability in belief. $\Sigma^{\text{belief}}$ is the rank-one component in the $\mathbf{f}'\mathbf{f}'^\top$ direction due to feedback of variable beliefs, and $\alpha$ is the proportionality constant from (5).

## Mathematical details for task-independent noise

We consider three potential sources of task-independent noise in posteriors: first, there are additional "high level" variables in $\mathbf{I}$ that may be probabilistically related to $\mathbf{x}$ but are not task-relevant. Just as variability in $\pi$ induces variability in $p_b(\mathbf{x}|\mathbf{E}(s), \pi)$, variability in these other internal states may induce variability in the posterior. Second, there may be measurement

noise in the observation of $\mathbf{E}$ or noise in the neurons afferent to those representing $\mathbf{x}$, resulting in a variable likelihood function even for constant stimuli [83]. Third, the stimulus itself may be stochastic by design, drawn according to some $p_e(\mathbf{E}|s)$, which we reparameterize below as a deterministic function of both $s$ and the noise, $\mathbf{E}(s, \boldsymbol{\epsilon}_\mathbf{E})$. In our notation, we partition the complete noise in the posterior, $\boldsymbol{\epsilon}$, into $\{\boldsymbol{\epsilon}_\mathbf{I}, \boldsymbol{\epsilon}_L, \boldsymbol{\epsilon}_\mathbf{E}\}$ corresponding to "internal state" noise, "likelihood" noise, and stimulus noise respectively.

We assume that the all noise sources are unaffected by task learning or task context and are independent of both $s$ and $\pi$.

By approximating the joint effect of $\pi$ and $\boldsymbol{\epsilon}_\mathbf{I}$ on the density of $\mathbf{x}$ as multiplicative, the full posterior decomposes as follows:

$$p_b(\mathbf{x}|s, \pi; \epsilon) = \frac{p_b(\mathbf{E}(s, \epsilon_\mathbf{E})|\mathbf{x}; \epsilon_L)p_b(\mathbf{x}|\epsilon_\mathbf{I}, \pi)p_b(\epsilon_\mathbf{I})p_b(\pi)}{p(s, \pi)p(\epsilon)}$$
$$\propto \underbrace{p_b(\mathbf{E}(s, \epsilon_\mathbf{E})|\mathbf{x}; \epsilon_L)}_{(i)}\underbrace{p_b(\mathbf{x}|\pi)}_{(ii)}\underbrace{p_b(\mathbf{x}; \epsilon_\mathbf{I})}_{(iii)}.$$

The first term ($i$) is the "noisy likelihood" conditioned on the noisy stimulus $\mathbf{E}(s, \boldsymbol{\epsilon}_\mathbf{E})$. The second term ($ii$) is the task-dependent component of the prior studied above. The third term ($iii$) captures the influence due to other internal variables besides $\pi$.

The two noise terms, ($i$) and ($iii$), may be combined into a single term. With some slight abuse of notation, we can replace $p_b(\mathbf{E}(s, \epsilon_\mathbf{E})|\mathbf{x}; \epsilon_L)$ with $p_b(s|\mathbf{x}; \epsilon_L, \epsilon_\mathbf{E})$ so that the $\boldsymbol{\epsilon}$ terms appear together. Combining terms, one can thus interpret both ($iii$) and ($i$) as noise in the likelihood, despite one arising from feedback and the other being feed-forward:

$$p_b(\mathbf{x}|s, \pi; \epsilon) \propto \overbrace{p_b(s|\mathbf{x}; \epsilon_L, \epsilon_\mathbf{E})p_b(\mathbf{x}; \epsilon_\mathbf{I})}^{(i),(iii)}\overbrace{p_b(\mathbf{x}|\pi)}^{(ii)}$$
$$\propto p_b(s|\mathbf{x}; \epsilon)p_b(\mathbf{x}|\pi).$$

This motivates our discussion only of "noisy likelihoods" in the main text—it implicitly includes stimulus noise, feedforward noise, and noise due to variable internal states besides $\pi$.

**Definition of linear Distributional Codes (LDCs).**  We define a Linear Distributional Code (LDC) as any distributional code with the property that the spike count of neurons encoding a *mixture* of two distributions is equal to the corresponding average of spike counts (note that this is a weaker condition than requiring the same to hold for all statistics of $\mathbf{r}$, so a bimodal p need not be encoded with a bimodal neural response distribution). Formally, let $p_\alpha(\mathbf{x})$ be a mixture of two arbitrary distributions, $\alpha p_1(\mathbf{x}) + (1 - \alpha)p_2(\mathbf{x})$, $0 \leq \alpha \leq 1$. We define "Linear Distributional Codes" (LDCs) as those codes where the following property holds for all neurons $i$:

$$\mathbb{E}_{\mathbf{r}\sim\mathcal{R}[p_\alpha(\mathbf{x})]}[S_i(\mathbf{r})] = \alpha\mathbb{E}_{\mathbf{r}\sim\mathcal{R}[p_1(\mathbf{x})]}[S_i(\mathbf{r})] + (1 - \alpha)\mathbb{E}_{\mathbf{r}\sim\mathcal{R}[p_2(\mathbf{x})]}[S_i(\mathbf{r})]. \tag{23}$$

Recall that $S_i(\mathbf{r})$ was defined earlier as the function that simply counts spikes of neuron $i$. Stated simply, (23) requires that the average firing rate encoding a mixture of distributions is equal to the mixture of the two firing rates for each individual distribution.

LDCs have the property that average firing rates are unaffected by averaging over noise. This is because if firing rates are linear for mixtures of any two distributions as in (23), then they will be linear in mixtures of three, four, or any number of distributions by simply applying (23) recursively (summing distributions is associative). Below, we will make use of this by

passing noise through the firing rate as follows:

$$\mathbb{E}_{\mathbf{r} \sim \mathcal{R}[\int_{\epsilon} \mathrm{p}(\mathbf{x}; \epsilon) \mathrm{d}\epsilon]}[S_i(\mathbf{r})] = \mathbb{E}_{\mathrm{p}(\epsilon)}[\mathbb{E}_{\mathbf{r} \sim \mathcal{R}[\mathrm{p}(\mathbf{x}; \epsilon)]}[S_i(\mathbf{r})]] .\tag{24}$$

On the left is the firing rate response to the marginal $\mathrm{p}(\mathbf{x})$, and on the right is the average firing response across noise draws. By definition, this expression only holds *in general* for LDCs, but note that a nonlinear distributional code may still appear "locally" linear in a particular task with a particular kind of noise.

**Extending results on variable beliefs to non-negligible noise, applying LDCs.** In what follows, expressions which do not explicitly contain "$\epsilon$" should be understood as the marginal distribution over $\mathbf{x}$ with noise averaged out. For instance, Eq (2) in the main text is unchanged: optimal learning in the presence of noise again implies a prior that is equal to the average of (noisy) posteriors seen in the task. Explicitly,

$$\mathrm{p}_b(\mathbf{x}|C = c) = \mathbb{E}_{\epsilon}[\mathbb{E}_{\mathrm{p}_e(s|C=c)}[\mathrm{p}_b(\mathbf{x}|s; \epsilon)]] ,$$

which implies a prior conditioned on graded beliefs $\pi$ of the form

$$\mathrm{p}_b(\mathbf{x}|\pi) = \mathbb{E}_{\epsilon}[\pi \mathbb{E}_{\mathrm{p}_e(s|C=2)}[\mathrm{p}_b(\mathbf{x}|s; \epsilon)] + (1 - \pi)\mathbb{E}_{\mathrm{p}_e(s|C=1)}[\mathrm{p}_b(\mathbf{x}|s; \epsilon)]] ,\tag{25}$$

which is identical to (3) if $\mathrm{p}(\mathbf{x}|s)$ is likewise understood to be marginalized over $\epsilon$. Thus, the introduction of noise does not affect either $\frac{\mathrm{d}\mathrm{p}_b(\mathbf{x}|\ldots)}{\mathrm{d}\pi}$ nor, by extension, $\frac{\mathrm{d}f}{\mathrm{d}\pi}$, at least in terms of notation, but should now be understood as containing an inner marginalization step over $\epsilon$.

Tuning with respect to $s$, on the other hand, will in general be altered by noise. Let $f_i(s, \pi, \epsilon)$ denote the expected spike count for neuron $i$ given a fixed stimulus $s$, belief $\pi$, and noise $\epsilon$ (it is still an *expected* spike count because of remaining stochasticity in $\mathcal{R}$). As before, we fix $\pi = {}^1/_2$ and study the changes in $f_i$ in response to $\pm\Delta s$:

$$\frac{\mathrm{d}f_i}{\mathrm{d}s} \approx \frac{1}{2\Delta s}\mathbb{E}_{\epsilon}[f_i(+\Delta s, {}^1/_2, \epsilon) - f_i(-\Delta s, {}^1/_2, \epsilon)] .$$

Crucially, the noise term $\epsilon$ appears as an outer expectation, since on each individual trial at $s = \pm\Delta s$ the noise will manifest differently. Expanding the definition of $f_i$ and applying the linearity property of LDCs in (24), we get

$$\frac{\mathrm{d}f_i}{\mathrm{d}s} \approx \frac{1}{2\Delta s}\left(\mathbb{E}_{\mathbf{r} \sim \mathcal{R}[\mathrm{p}(\mathbf{x}|+\Delta s; \pi={}^1/_2)]}[S_i(\mathbf{r})] - \mathbb{E}_{\mathbf{r} \sim \mathcal{R}[\mathrm{p}(\mathbf{x}|-\Delta s; \pi={}^1/_2)]}[S_i(\mathbf{r})]\right) .\tag{26}$$

The LDC assumption allows the outer expectation over $\epsilon$ to be pushed into an expectation inside of the $\mathcal{R}[\ldots]$, which is equivalent to marginalizing the posterior $\mathrm{p}(\mathbf{x}| \pm \Delta s, {}^1/_2)$ over $\epsilon$.

Note that by marginalizing out the noise, both (25) and (26) can be written without an explicit $\epsilon$, so the remaining proof becomes identical to the proof of (5) given above. This implies that in cases with significant task-independent noise, LDCs will have the property that $\frac{\mathrm{d}f}{\mathrm{d}s} \widetilde{\propto} \frac{\mathrm{d}f}{\mathrm{d}\pi}$, and hence make all the same predictions for data described in the main text, such as the emergence of both differential correlations and a top-down component of choice probabilities proportional to neural sensitivities over learning. While the LDC assumption is *sufficient* to guarantee our initial results generalize to cases with non-negligible noise in all contexts, it is not strictly *necessary* in any one context since a nonlinear distributional code may appear linear in a particular context.

**Derivation of (10): Filtering noise by a static self-consistent prior.** Throughout this section, we will fix $s = 0$ and $\pi = {}^1/_2$ to isolate the effects of $\epsilon$ in "zero-signal" conditions. We will

also assume that $\mathbf{x}$ is discrete so that we can use finite-length vectors of probability mass rather than probability density functions, but this is only for intuition and notational convenience.

Above, we used the chain rule of derivatives to write neurons' sensitivity to various factors in terms of their sensitivity to the posterior density, $d\mathbf{f}/dp_b(\mathbf{x}|\ldots)$. To a first approximation, the same trick can be applied to write the *covariance* of neural responses in terms of their sensitivity to $p_b(\mathbf{x}|\ldots)$ and the *covariance* in the posterior mass itself due to task-independent noise ($\boldsymbol{\epsilon}$). To make use of vector notation and linear algebra intuitions, let $\mathbf{p} \equiv p(\mathbf{x}|\ldots)$ denote the vector of probability mass over a discrete $\mathbf{x}$. Then, to a first approximation, the covariance between neurons $i$ and $j$ is

$$\Sigma_{ij}^{\boldsymbol{\epsilon}} \approx \nabla_{\mathbf{p}} f_i^\top \Sigma_{\mathbf{p}} \nabla_{\mathbf{p}} f_j. \tag{27}$$

The inner term, $\Sigma_{\mathbf{p}}$, is the covariance of the elements of the posterior $\mathbf{p}$—i.e. how the probability mass at pairs of points $\mathbf{x}_1$ and $\mathbf{x}_2$ (co)varies due to $\boldsymbol{\epsilon}$ (see S1 Text for further discussion of this term). The term $\nabla_{\mathbf{p}} f_i$ is the gradient of neuron $i$'s firing rate with respect to first-order changes in the probability mass $\mathbf{p}$.

Recall that the noisy posterior, $p_b(\mathbf{x}|s, \pi; \boldsymbol{\epsilon})$, can be written with all noise terms in the likelihood, i.e. $p_b(\mathbf{x}|\pi)p_b(s|\mathbf{x}; \boldsymbol{\epsilon})$ (up to constants). Because of this, the prior may be pulled out of $\Sigma_{\mathbf{p}}$ as follows (we drop $\pi = {}^1/_2$ here to reduce clutter):

$$\begin{aligned}
\Sigma_{\mathbf{p}}(\mathbf{x}_1, \mathbf{x}_2) \quad &= \mathbb{E}_{\epsilon}[(p_b(\mathbf{x}_1|s=0; \epsilon) - \mathbb{E}_{\epsilon'}[p_b(\mathbf{x}_1|s=0; \epsilon')]) \\
&\qquad \times (p_b(\mathbf{x}_2|s=0; \epsilon) - \mathbb{E}_{\epsilon'}[p_b(\mathbf{x}_2|s=0; \epsilon')])] \\
&\propto \mathbb{E}_{\epsilon}[(p_b(\mathbf{x}_1)p_b(s=0|\mathbf{x}_1; \epsilon) - \mathbb{E}_{\epsilon'}[p_b(\mathbf{x}_1)p_b(s=0|\mathbf{x}; \epsilon')]) \\
&\qquad \times (p_b(\mathbf{x}_2)p_b(s=0|\mathbf{x}_2; \epsilon) - \mathbb{E}_{\epsilon'}[p_b(\mathbf{x}_2)p_b(s=0|\mathbf{x}; \epsilon')])] \\
&= p_b(\mathbf{x}_1)p_b(\mathbf{x}_2)\Sigma_{\mathbf{p}}^{LH}(\mathbf{x}_1, \mathbf{x}_2),
\end{aligned}$$

where

$$\begin{aligned}
\Sigma_{\mathbf{p}}^{LH}(\mathbf{x}_1, \mathbf{x}_2) \quad &\equiv \mathbb{E}_{\epsilon}[(p_b(s=0|\mathbf{x}_1; \epsilon) - \mathbb{E}_{\epsilon'}[p_b(s=0|\mathbf{x}; \epsilon')]) \\
&\qquad \times (p_b(s=0|\mathbf{x}_2; \epsilon) - \mathbb{E}_{\epsilon'}[p_b(s=0|\mathbf{x}; \epsilon')])].
\end{aligned}$$

In the second line, we absorbed $p_b(s=0)$ terms into a proportionality constant since we are primarily interested in the shape of $\Sigma_{\mathbf{p}}$, or its relative rather than absolute eigenvalues. This can be rewritten in matrix notation as

$$\Sigma_{\mathbf{p}} \propto \text{diag}(p_b(\mathbf{x}))\Sigma_{\mathbf{p}}^{LH}\text{diag}(p_b(\mathbf{x})), \tag{(10)restated}$$

where $\Sigma_{\mathbf{p}}^{LH}$ is the covariance *of the likelihood* with $s = 0$ and is task-independent. The prior, $p_b(\mathbf{x}|\pi = {}^1/_2))$, is task-dependent. Eq (10) thus gives, to a first approximation, an expression for how noise in the likelihood is sculpted by learning: the "intrinsic" covariance in the likelihood, which is present even before learning, is pre- and post-multiplied by a diagonal matrix of the task-dependent prior mass vector.

One way to reason about (10) is by considering its eigenvector decomposition. For instance, *differential correlations* are introduced to the extent that the relative variance in the $\frac{dp_b}{ds}$ direction is increased after left- and right-multiplying the intrinsic noise ($\Sigma_{\mathbf{p}}^{LH}$) by the diagonal matrix of prior probabilities. It is nontrivial, however, to state this in terms of conditions on $\mathbf{x}$, $s$, or $\mathcal{R}$, which we leave as a problem for future work.

## Numerical details for Fig 4: Filtering noise by a static self-consistent prior

Fig 4 was created by simulating a discretized 2D space with both $x$ and $y$ coordinates ranging in $[-5, 5]$ for both $\mathbf{E}$ and $\mathbf{x}$. We set the distribution of observations, $p_e(\mathbf{E}|s)$, to be a Gaussian whose mean or covariance depended on $s$. One value of $\mathbf{E}$ was sampled from this Gaussian on each "trial" and provided to the learning model as an observation. In Fig 4A–4K, the mean of $p_e(\mathbf{E}|s)$ was parameterized by a smooth (cubic) function of $s$,

$$\mu_1(s) = s, \qquad \mu_2(s) = (s + s^3)/10,$$

and the covariance of $p_e(\mathbf{E}|s)$ was set to a constant $\begin{bmatrix} 0.5 & 0 \\ 0 & 0.5 \end{bmatrix}$. For Fig 4L–4V, the mean of $p_e(\mathbf{E}|s)$ was set to zero while the covariance of $p_e(\mathbf{E}|s)$ was parameterized by $s$ as follows:

$$\Sigma = \begin{cases} \Sigma^0 + \tanh(s)\Sigma^+ & \text{if } s > 0 \\ \Sigma^0 - \tanh(s)\Sigma^- & \text{if } s \leq 0 \end{cases}$$

where $\Sigma^0 = \begin{bmatrix} 0.5 & 0 \\ 0 & 0.5 \end{bmatrix}$, $\Sigma^+ = \begin{bmatrix} 1 & 1 \\ 1 & 1 \end{bmatrix}$, and $\Sigma^- = \begin{bmatrix} 1 & -1 \\ -1 & 1 \end{bmatrix}$. In all cases, $p_e(s)$ was set to a uniform distribution in $[-3, +3]$.

In addition to the two different distributions on $\mathbf{E}$ just defined, we considered two cases for the brain's internal model—one in which the generative model $p_b(\mathbf{E}|\mathbf{x})$ is assumed to be precise, and one in which it is assumed to be imprecise. In both cases, we assumed $p_b(\mathbf{E}|\mathbf{x}) = \mathcal{N}(\mathbf{E}; \mathbf{x}, \Sigma_{\mathbf{E}})$, i.e. that $\mathbf{E}$ is Gaussian-distributed, centered at the corresponding value of $\mathbf{x}$. In the precise case, we set $\Sigma_{\mathbf{E}} = \begin{bmatrix} 0.04 & 0 \\ 0 & 0.04 \end{bmatrix}$, and in the imprecise case we set $\Sigma_{\mathbf{E}} = \begin{bmatrix} 0.36 & 0 \\ 0 & 0.36 \end{bmatrix}$. These parameters were chosen so that $p_b(\mathbf{E}|\mathbf{x})$ was always narrower than the marginal distribution of evidence $p_e(\mathbf{E})$, which ensured that a prior $p_b(\mathbf{x})$ could be found such that the marginal distribution on $\mathbf{E}$ is correct, or $p_b(\mathbf{E}) = \int_{\mathbf{x}} p_b(\mathbf{x}) p_b(\mathbf{E}|\mathbf{x}) = p_e(\mathbf{E})$.

We are interested in adapting the brain's prior, $p_b(\mathbf{x})$, to the marginal statistics of observations defined by the task, or $p_e(\mathbf{E}) = \int_s p_e(s) p_e(\mathbf{E}|_s) ds$. To achieve this, we discretized $\mathbf{x}$ space so that $p_b(\mathbf{x})$ is a high-dimensional vector, and optimized the following objective by gradient descent:

$$\text{Loss} \equiv \text{KL}(p_e(\mathbf{E})||p_b(\mathbf{E})) - \lambda \mathcal{H}(p_b(\mathbf{x})), \tag{28}$$

where $\mathcal{H}(p_b(\mathbf{x}))$ is the entropy of the prior, included for regularization, and $\lambda$ sets its importance; we used $\lambda = 0.001$ throughout.

Gradient descent in this space is both more stable and simpler to implement using the log prior rather than the prior. Changing variables to the log-prior simply requires multiplying the Loss gradient by the prior itself, since $\nabla_{\log p_b(\mathbf{x})} \text{Loss} = p_b(\mathbf{x}) \nabla_{p_b(\mathbf{x})} \text{Loss}$. The gradient of the KL term in (28), with respect to the prior density $p_b(\mathbf{x})$, is given by

$$\nabla_{p_b(\mathbf{x})} \text{KL}(p_e(\mathbf{E})||p_b(\mathbf{E})) = -\frac{1}{p_b(\mathbf{x})} \mathbb{E}_{p_e(\mathbf{E})} \left[ p_b(\mathbf{x}|\mathbf{E}) \right],$$

which after the log-transform simplifies to

$$\nabla_{\log p_b(\mathbf{x})} \mathrm{KL}(p_e(\mathbf{E}) || p_b(\mathbf{E})) = -\mathbb{E}_{p_e(\mathbf{E})}[p_b(\mathbf{x}|\mathbf{E})] \,. \tag{29}$$

Remarkably, (29) means that updating the prior in the direction that maximally improves the match to the marginal distribution of observations simply requires taking a step with the *log prior* in the direction of the *average posterior*. Next, the gradient of $\mathcal{H}(p_b(\mathbf{x}))$ with respect to the log prior is

$$\nabla_{\log p_b(\mathbf{x})} \mathcal{H}(p_b(\mathbf{x})) = -p_b(\mathbf{x})(1 + \log p_b(\mathbf{x})) \,, \tag{30}$$

giving a combined gradient of

$$\nabla_{\log p_b(\mathbf{x})} \mathrm{Loss} = -\mathbb{E}_{p_e(\mathbf{E})}[p_b(\mathbf{x}|\mathbf{E})] + \lambda p_b(\mathbf{x})(1 + \log p_b(\mathbf{x})) \,. \tag{31}$$

Starting with a uniform prior over the discretized 2D space of $\mathbf{x}$ values, learning consisted of drawing a large number of random likelihoods (randomizing both $s$ and $\mathbf{E}$) to estimate the average posterior, then the log prior was updated by taking a gradient descent step along (31) while ensuring the constraint that $\int_{\mathbf{x}} p_b(\mathbf{x}) d\mathbf{x} = 1$ after each iteration. We found that a large initial stepsize of 100 was required for learning, which we then halved every hundred iterations for a total of 300 iterations.

To measure the change in covariance of the posterior density itself along $dp_b(\mathbf{x}|\ldots)/ds$, we compared the first and last iteration, which have the same set of variable likelihoods (i.e. identical $p_e(\mathbf{E}|s)$) but different priors. For each, we drew a large set of $\mathbf{E} \sim p_e(\mathbf{E}|s = 0)$, and numerically computed the covariance of the posterior mass on $\mathbf{x}$ in the discretized space, $\Sigma_p$. Because we had discretized $\mathbf{x}$ into 85 bins in each dimension, this makes $\Sigma_p$ a $7225 \times 7225$ matrix. We likewise computed $dp_b(\mathbf{x}|\ldots)/ds$ separately before and after learning (Fig 4E, 4J, 4P and 4U show $dp_b(\mathbf{x}|\ldots)/ds$ after learning) by drawing a large number of random posteriors and taking the difference of their average at $s = +.01$ and $s = -.01$. We plotted the change in relative variance along $dp_b(\mathbf{x}|\ldots)/ds$ in Fig 4F, 4K, 4Q and 4V, defined as

$$\frac{\mathbf{u}^{\top} \Sigma_p \mathbf{u}}{\mathrm{Trace}(\Sigma_p)} \,,$$

where $\mathbf{u}$ is the unit vector pointing in the $dp_b(\mathbf{x}|\ldots)/ds$-direction.

This entire process—including learning and estimating the fraction of variance along the $dp_b(\mathbf{x}|\ldots)/ds$ direction—was repeated 4 times for each experiment configuration. We found that these results were highly repeatable; error bars in Fig 4F, 4K, 4Q and 4V indicate standard error of the mean across the 4 runs of each condition.

## Numerical details for Fig 6: Inferring the internal model

Complex tasks (e.g. those switching between different contexts), or incomplete learning (e.g. uncertainty about fixed task parameters), will often induce variability in multiple internal beliefs about the stimulus. Assuming that this variability is independent between the beliefs, we can write the observed covariance as $\Sigma \approx \Sigma^0 + \sum_k \lambda^{(k)} \mathbf{u}^{(k)} \mathbf{u}^{(k)\top}$. Here, each vector $\mathbf{u}^{(k)}$ corresponds to the change in the population response corresponding to a change in internal belief $k$. The coefficients $\lambda^{(k)}$ are proportional to the variance of the trial-to-trial variability in belief $k$, as in $\mathrm{var}(\pi)$ above, and $\Sigma^0$ represents all task-independent covariance.

The model in our proof-of-concept simulations has been described previously [34]. In brief, it performs inference by neural sampling in a linear sparse-coding model [3, 48, 72]. The prior is derived from an orientation discrimination task with two contexts—oblique

orientations and cardinal orientations—that is modeled on an analog direction discrimination task [22]. We simulated the responses of 1024 V1 neurons whose receptive fields uniformly tiled the orientation space. Each neuron's response corresponds a set of samples from the posterior distribution over the intensity of its receptive field in the input image. We simulated zero-signal trials by presenting white noise images to the model. The eigenvectors not described in the main text correspond to stimulus-driven covariability, plotted in S3 Fig for comparison.

## Supporting information

**S1 Fig. Visualizing the limiting process(es) of stimulus distributions as defined by Eqs (18) and (19). a)** Initially, the distribution on stimuli may be wide, here illustrated as a Gaussian that is split by the two categories. **b)** Eq (18) considers the case where *each* category goes to a Dirac delta around some $\pm\Delta s$, plus a delta at zero. **c)** As the magnitude of $\Delta s$ gets small, the approximation in (5) gets better. As discussed in the methods, this limit may not be taken fully to $\Delta s \to 0$.
(EPS)

**S2 Fig. Gaussian-mixture demonstration of the limiting process(es) of stimulus distributions. a)** Simple generative model simulated in **b-d**. $x$ is a scalar drawn from a Gaussian around $\pm\mu_x$ (matching the sign of $C$), and the stimulus $s$ is drawn from a Gaussian around $x$. **b)** The prior on $x$ is a mixture of two Gaussians. Colors correspond to different values of $\mu_x$. **c)** Derivatives of the posterior with respect to $s$. **d)** Derivatives of the posterior with respect to $\pi$. The match to **c** improves as $\mu_x$ gets closer to 0, which simulates changes to the learned model as stimulus categories $\mu_x$ draw closer together (as in S1c Fig).
(EPS)

**S3 Fig. Principal components of model neurons due to only stimulus-driven correlations.** Note that the sinusoidal eigenvectors at the same frequency have indistinguishable eigenvalues and hence form quadrature pairs, implying circular symmetry with respect to neurons' tuning. There is no more variance along the vertical-horizontal preferred orientation axis than then oblique axis.
(EPS)

**S1 Text. Supplemental text.**
(PDF)

## Acknowledgments

We thank the many colleagues with whom we have discussed aspects of this work and who have provided us with valuable feedback, in particular (alphabetically) Matthias Bethge, Adrian Bondy, Bruce Cumming, Alex Ecker, Jakob Macke, Ruben Moreno-Bote, Hendrikje Nienborg, and Emmett Wyman.

## Author Contributions

**Conceptualization:** Richard D. Lange, Ralf M. Haefner.

**Formal analysis:** Richard D. Lange, Ralf M. Haefner.

**Software:** Richard D. Lange, Ralf M. Haefner.

**Supervision:** Ralf M. Haefner.

**Visualization:** Richard D. Lange.

**Writing – original draft:** Richard D. Lange, Ralf M. Haefner.

**Writing – review & editing:** Richard D. Lange, Ralf M. Haefner.

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
