## [Decision Letter · Decision Letter 0]

10 Aug 2021

Dear Dr. Lange,

Thank you very much for submitting your manuscript "Task-induced neural covariability as a signature of approximate Bayesian learning and inference" for consideration at PLOS Computational Biology. As with all papers reviewed by the journal, your manuscript was reviewed by members of the editorial board and by several independent reviewers. The reviewers appreciated the attention to an important topic. Based on the reviews, we are likely to accept this manuscript for publication, providing that you modify the manuscript according to the review recommendations.

Sincerely,

Ming Bo Cai

Associate Editor

PLOS Computational Biology

Wolfgang Einhäuser

Deputy Editor

PLOS Computational Biology

[LINK]

Reviewer's Responses to Questions

**Comments to the Authors:**

Reviewer #1: REVIEW for “Task-induced neural covariability as a signature of approximate Bayesian learning and inference” by Lange and Haefner.

This paper analytically derived neural signature of tasks-specific signature by analyzing noise correlation and choice probability for 2-AFC task in a Bayesian probabilistic coding framework. The analytical results are supplemented by simulations when closed-form solutions were difficult to obtain. The paper also summarized previous experimental results that could be interpreted as evidence supporting the theory. Furthermore, the authors used a simple principal component analysis (PCA) method to show that the statistical structure of the neural responses could be used to infer internal belief about the stimulus in simulated/synthetic data.

Overall, I find this paper to be a valuable contribution to the field of neural coding. It contains various interesting ideas, which are likely to have an impact on the interpretation of noise correlations in the context of feedforward v.s. feedback processing. The analytical derivation under the Bayesian framework is elegant and thorough. There are some concerns I have with regard to the writing, along with some technical points, which need to be further clarified. Although these concerns slightly reduce the enthusiasm, I still consider this paper as a useful and timely addition to the literature.

One main concern of the current version is the readability. The paper is bit difficult to understand at this point. This is partly due to that many results were presented, but there seems to be a lack of focus.

i) The predictions are scattered throughout the paper. The authors may consider summarizing the predictions explicitly somewhere, perhaps in the beginning of the Discussion.

ii) The first three sections of the Results are basically setting up the stages without actual results. I found them to be difficult to follow because too many concepts were introduced/defined. It would be useful if these could be streamlined and compressed.

Another more technical concern is the use of the term “differential correlation” which is used many times in the paper. In the original publication by Moreno-Bote et al (2014), “differential correlation” was defined based on the derivative of the tuning curve at each stimulus value. However, in the present paper, “differential correlation” is computed based on the difference between a pair of stimuli, i.e., f(\\theta_1) - f(\\theta_2)=\\delta f. Although some recent studies have been making no distinction about the two, this does not seem to be a trivial distinction, because Fisher information at any stimulus value is related to the derivative of the tuning curve— this is defined in the absolute sense for that particular stimulus, not relative to another stimulus. Thus fundamentally, it is related to local discrimination, not general discrimination.

Relatedly, there is a sense that the noise correlation along the f’direction ( f’ represents the derivative of the tuning curve, not \\delta f for an arbitrary pair) is specifical, because a small input noise (or stimulus noise) will naturally lead to a component in this direction. Therefore, I find the arguments around Fig. 6c to be unconvincing. In particular, the statement in lIne 449-452 [“This makes the f′ direction largely arbitrary, so the results of Rabinowitz et al. (2015), as well as other recent findings of alignment between f′ and the low-rank modes of covariance (Bondy et al., 2018; Montijn et al., 451 2019; Rumyantsev et al., 2020), ought to be extremely surprising in a feedforward framework or in models of low-rank but task-independent variability”] is quite problematic in my view due to the impact of shared input noise. The same statement was basically repeated in line 552-554. I don’t see why this should be considered as “extremely surprising” as claimed in the paper. In general, statements like this are perhaps too subjective to be useful.

Regarding the generality of the results: the results in the paper are limited to the 2-AFC task. The continuous estimation problem is only briefly discussed in SI (line 1259-1273). It would be useful to bring the discussions on continuous estimation problem up to the main text. This will help partly address the question of generality. Also, Fisher information (and differential correlation) comes naturally with the continuous estimation problem.

Specific comments:

* Line 19-20. It is stated that the predictions of the proposed framework differ from the ones form the feedforward models. Reading through the paper, I am not exactly sure what these differences are. It would be useful to summarize these explicitly.

* Line 39-41 stated that the results hold under general assumptions. However, later it was shown that, when there is noise, the results only hold in some cases. There appears to be a discrepancy between what was claimed here (as well as lines 118-120) and the actual results. One of proof is based on taking the limit in another limiting regime, which is delicate. It would be helpful to re-write these sentences to reflect the subtlety of the results.

* Could the authors clearly state the assumptions underlying the “fundamental self-consistency relationship”, i.e. “the average posterior equals to the prior”? An average reader might not be familiar with this relationship.

* “differential correlation” needs to be explicitly and clearly defined (also see my comments earlier).

*Line 534-536 stated that “some” of the measured “differential” correlation could be understood as near-optimal feedback. I found this description to be rather vague. What if the feedback is not near-optimal? In that case, would some “differential” correlation still be predicted? If the answer is yes, then the interpretation of “near-optimal feedback” based on “differential” correlation would not be valid.

* Line 522-524. The connection to the recurrent neural networks is not clearly described.

* For the clarity of the presentation, it would be useful to label the subsections of Method for the sake of referencing in the main text. Currently, it is difficult to locate the proof for a particular statement stated in the main text.

* Would it be useful to fit a regression line in Fig. 6a? That might help better convey the point.

* It is difficult to see the connection between Fig. 6b and the claims made in Line 439-443, and why Fig 6b is supporting the authors’ claim. Could the authors please unpack the connection there?

* Line 334-349 describes a special class of encoding schemes for which the results (Eq. 8 and Eq. 9) still hold when the noise is present. Is this class of code both necessary and sufficient? It is obviously sufficient based on what is described there, but unclear whether it is also necessary. For the prediction in Line 347-349, I’d think one would need the linearity to be a necessary condition as well. If it is both necessary and sufficient, it would be useful to state that explicitly. If it is not, it would be useful to tone down the claim in the last sentence.

*Line 774-775 is difficult to understand. “map” or “mapping”?

*Line 182- 187. These lines are difficult to follow.

*Fig 1d. I found the scheme is confusing, in particular the direction of arrows. Maybe I don’t quite understand what this figure is trying to show.

*Line 110: “two frameworks”. Not sure if it would be accurate to call the studies on noise correlation as a “framework”—perhaps more fair to use “two branches of research”?

Reviewer #2: The manuscript discusses the effect of probabilistic inference on neural response statistics. In particular, the focus of the paper is inference of task-related variables on the covariation of neural responses. The paper establishes normative relationship between distinct neural phenomena, such as top-down feedback, choice probabilities and information limiting correlations. Besides the appeal that a formal treatment is provided to higher order response statistics, a further aspect that helps appreciating the theory is that it discusses alternative computational approaches jointly. The paper provides important insights into the way computation-level considerations are linked to neural population activity.

The paper is very dense but is clearly written and arguments are well supported by formal analysis. I only have a number of minor clarification suggestions/questions.

Questions:

I have one question that might require some additional writing. In the paragraph starting at line 260 the authors lay out the requirements for their analytical treatment. The assumptions are sound and adequate for the particular setting they analyze but the authors would benefit from a discussion of how these assumptions affect the generality of the claims. For instance, while a fully learned model can be assumed but some qualitative aspects of the model would still hold even if there is a discrepancy between the internal model and the actual model of the task. Importantly, there are tools to infer variations in the strategy followed by animals from behavior alone, therefore one could expect these changes to be captured by the theory presented here. From an experimental point of view, the second assumption is certainly true for the analyzed experiments but the basic consequences of the theory are more general than stimuli presented close to the psychometric threshold.

In the section ‘Variable beliefs in the presence of noise’ the authors argue that the alignment of df/ds and df/d\\pi is critical for the derivations to hold and then they introduce LDC’s that can overcome theoretical hurdles. This fine theoretical insight is proposed to be a basis for experimental validation. While this is indeed true, additional insights would be welcome if this is something that can indeed be measured in electrophysiological data.

The authors argue that the self-consistency claim that the results are based on are general across theories of probabilistic computations. The argument for this claim is properly spelled out. It would be instructive for the reader to briefly discuss if these theories are indistinguishable in the discussed context.

clarifications:

line 110: I suggest spelling out the ‘two frameworks’: The reader might lose track by this time

Caption of fig 2: s is not defined in time to understand the figure.

Caption of fig 2: ‘Our derivation assumes that smoothly changing posteriors (a) corresponds to smooth changes in neural responses’ -> one could introduce f(s) here since later f’ turns up in the caption.

line 149: ‘variability of posterior distributions’ is not sufficiently motivated at this point

Caption of fig 3: ‘Variability in the underlying posterior may appear as correlated variability in spike counts.’ -> I appreciate the argument but I am not sure how the figure motivates this point.

line 255: please explain \\pi = \\frac{12}

line 274: Changes induced by top-down influences affect the tuning curve. It would be useful to note how this affects measuring tuning curves

**Have the authors made all data and (if applicable) computational code underlying the findings in their manuscript fully available?**

Reviewer #1: **No: **The authors stated that "No new data were collected in this work. Matlab code for simulation results will be made available on https://github.com/haefnerlab ."

Reviewer #2: Yes

PLOS authors have the option to publish the peer review history of their article (what does this mean?). If published, this will include your full peer review and any attached files.

Reviewer #1: No

Reviewer #2: No

Figure Files:

Data Requirements:

Reproducibility:

References:

---

## [Decision Letter · Decision Letter 1]

12 Oct 2021

Dear Dr. Lange,

We are pleased to inform you that your manuscript 'Task-induced neural covariability as a signature of approximate Bayesian learning and inference' has been provisionally accepted for publication in PLOS Computational Biology.

Best regards,

Ming Bo Cai

Associate Editor

PLOS Computational Biology

Wolfgang Einhäuser

Deputy Editor

PLOS Computational Biology

We are very happy to accept the paper for publication. Please include a link to the code in the manuscript when you finalize the print-ready paper. This does not need further review.

Reviewer's Responses to Questions

**Comments to the Authors:**

Reviewer #1: I thank the authors for their response to my questions. The revised version is substantially improved. I would like to recommend the publication of this paper. I consider that it will be a useful contribution to the field of neural coding.

Reviewer #2: The authors did an excellent job and I believe that the paper is ripe for publication.

**Have the authors made all data and (if applicable) computational code underlying the findings in their manuscript fully available?**

Reviewer #1: **No: **The authors promise to release the code upon publication of this paper.

Reviewer #2: Yes

PLOS authors have the option to publish the peer review history of their article (what does this mean?). If published, this will include your full peer review and any attached files.

Reviewer #1: No

Reviewer #2: No

---

## [Editor Report · Acceptance letter]

2 Mar 2022

PCOMPBIOL-D-21-00963R1 

Task-induced neural covariability as a signature of approximate Bayesian learning and inference

Dear Dr Lange,

I am pleased to inform you that your manuscript has been formally accepted for publication in PLOS Computational Biology. Your manuscript is now with our production department and you will be notified of the publication date in due course.

With kind regards,

Livia Horvath
